# The role of activated androgen receptor in cofilin phospho-regulation depends on the molecular subtype of TNBC cell line and actin assembly dynamics

**Lubna Tahtamouni**[1,2]*, **Ahmad Alzghoul**[1], **Sydney Alderfer**[3], **Jiangyu Sun**[2], **Mamoun Ahram**[4], **Ashok Prasad**[3], **James Bamburg**[2]

**1** Department of Biology and Biotechnology, Faculty of Science, The Hashemite University, Zarqa, Jordan, **2** Department of Biochemistry and Molecular Biology, College of Natural Sciences, Colorado State University, Fort Collins, CO, United States of America, **3** Department of Chemical and Biological Engineering, School of Biomedical Engineering, Colorado State University, Fort Collins, CO, United States of America, **4** Department of Physiology and Biochemistry, School of Medicine, The University of Jordan, Amman, Jordan

* lubnatahtamuni@hu.edu.jo, lubna.tahtamouni@colostate.edu

**Data Availability Statement:** The data underlying the results presented in the study are available as Supporting Information files.

## Abstract

Triple negative breast cancer (TNBC) is highly metastatic and of poor prognosis. Metastasis involves coordinated actin filament dynamics mediated by cofilin and associated proteins. Activated androgen receptor (AR) is believed to contribute to TNBC tumorigenesis. Our current work studied roles of activated AR and cofilin phospho-regulation during migration of three AR+ TNBC cell lines to determine if altered cofilin regulation can explain their migratory differences. Untreated or AR agonist-treated BT549, MDA-MB-453, and SUM159PT cells were compared to cells silenced for cofilin (KD) or AR expression/function (bicalutamide). Cofilin-1 was found to be the only ADF/cofilin isoform expressed in each TNBC line. Despite a significant increase in cofilin kinase caused by androgens, the ratio of cofilin:p-cofilin (1:1) did not change in SUM159PT cells. BT549 and MDA-MB-453 cells contain high p-cofilin levels which underwent androgen-induced dephosphorylation through increased cofilin phosphatase expression, but surprisingly maintain a leading-edge with high p-cofilin/total cofilin not found in SUM159PT cells. Androgens enhanced cell polarization in all lines, stimulated wound healing and transwell migration rates and increased *N/E-cadherin* mRNA ratios while reducing cell adhesion in BT549 and MDA-MB-453 cells. Cofilin KD negated androgen effects in MDA-MB-453 except for cell adhesion, while in BT549 cells it abrogated androgen-reduced cell adhesion. In SUM159PT cells, cofilin KD with and without androgens had similar effects in almost all processes studied. AR dependency of the processes were confirmed. In conclusion, cofilin regulation downstream of active AR is dependent on which actin-mediated process is being examined in addition to being cell line-specific. Although MDA-MB-453 cells demonstrated some control of cofilin through an AR-dependent mechanism, other AR-dependent pathways need to be further studied. Non-cofilin-dependent mechanisms that modulate migration of SUM159PT cells need to be investigated. Categorizing TNBC behavior as AR responsive and/or cofilin dependent can inform on decisions for therapeutic treatment.

**Funding:** Research was supported by generous gifts from family and friends through donations to the Colorado State University (CSU) Foundation, by research support grant from the College of Natural Sciences and the Department of Biochemistry and Molecular Biology (Grant# 1337455; LT) and CSU OVPR Research Continuity Award (1630620-OVPR; AP and LT). We gratefully acknowledge core infrastructure support for our microscope facilities and an NIH Shared Instrumentation Grant 1S10OD025127 (JB). The funders had no role in study design, data collection and analysis, decision to publish, or preparation of the manuscript.

**Competing interests:** The authors have declared that no competing interests exist.

# Introduction

Breast cancer is the most prevalent cancer among females; it accounts for almost 1 in 4 cancer cases [1]. Despite considerable advances in breast cancer therapy, metastasis remains the major cause of cancer-related deaths [2]. Breast cancer can be classified into major molecular subtypes according to the expression of estrogen receptor (ER), progesterone receptor (PR), and over-expression and/or amplification of the human epidermal growth factor 2 (HER2) [3–5]. Targeting ER, PR or HER2 in these cancers substantially improves disease-free survival [6, 7]. However, triple negative breast cancer (TNBC), a heterogeneous disease that constitutes 10–20% of all breast cancers, lacks the expression of these recognized therapeutic targets. Most metastatic TNBCs initially respond to adjuvant chemotherapy [8–10], however, most patients experience recurrence following treatment. Less than 30% of women with metastatic TNBC survive 5 years since it is difficult to treat, more aggressive, and has higher rates of distant recurrence than other breast cancer subtypes [4, 11, 12].

Gene expression and cluster analysis have been used to identify six subtypes of TNBCs including two basal-like (BL1 and BL2), mesenchymal-like (M), mesenchymal stem-like (MSL), immunomodulatory (IM), and luminal androgen receptor (LAR) subtype [11, 13]. Androgens such as testosterone and dihydrotestosterone (DHT) mediate their actions via the androgen receptor (AR); testosterone can be converted to its more biologically active form, DHT, by 5α reductase, and to estrogen by aromatase [14–16]. In mammary epithelial cells, AR, a ligand-dependent nuclear transcription factor, is the most abundant nuclear receptor [7]. AR binds androgens with strong affinity (low nanomolar) [17]. After binding DHT in the cytoplasm, AR gets released from chaperone proteins, homodimerizes, and translocates to the nucleus where it interacts with and directly controls the expression of androgen-response genes [18, 19].

Reorganization of the actin cytoskeleton is critical for various cellular processes such as cell division, formation of cell junctions, cell shape, and the migration and invasion of cancer cells [20, 21]. Assembly dynamics of the actin filament network at the leading edge as well as the cell rear allows cells to attach to a substratum, contract its body, push forward, and move [22, 23]. During metastasis, cells change from an epithelial phenotype to a mesenchymal phenotype exhibiting various migratory protrusions such as lamellipodia, filopodia, and invadopodia [24–26] a process known as epithelial-mesenchymal transition (EMT). EMT is dependent on the dynamic rearrangement of the actin cytoskeleton, which is regulated by actin binding proteins (ABP) [27, 28].

Different ABP establish and mediate dynamics of membrane protrusions at the leading edge of a normal migrating or metastatic cells [29, 30]. Cofilin, a filamentous actin (F-actin) binding and severing protein that belongs to the ADF/cofilin family of proteins, plays important roles in various stages of cancer progression including cell polarization and polarized migration [31–34], escape from apoptosis, and release of metalloproteases [35], all of which are important in the metastatic process. The activity of cofilin is controlled by different signaling pathways [36]. Cofilin can bind to membrane lipids such as phosphatidylinositol 4-phosphate (PIP) and phosphatidylinositol 4,5-bisphosphate (PIP2), which inhibits cofilin from binding to actin [37, 38]. In addition, the ability of cofilin to bind actin is inhibited by phosphorylation at Ser-3 by the LIM kinase (LIMK) family of serine/threonine kinases [39]. Cofilin phosphatases such as slingshot 1L (SSH-1L) and chronophin (CIN) counteract the phospho-inactivation of cofilin [40–43]. Cofilin mRNA has been reported to be increased in various malignant cells [44], and increased cofilin protein level has also been shown to be associated with poor prognosis in different carcinomas [45–47].

Several studies have explored the role of activated AR in breast cancer initiation and progression, and in the regulation of transcription of EMT-regulatory genes [48]. AR is expressed

in approximately 90% of ER+ breast cancers and its expression is related to a favorable prognosis by antagonizing ER [49, 50]. In TNBC, AR is expressed to some degree in 30–50% of tumors [11, 12]. It is generally accepted that activated AR stimulates cellular proliferation and spreading of ER-/AR+ TNBC [51–54]. In addition to TNBC, around 25% of breast cancer metastases express AR, whereas ER and PR levels are almost undetectable, but the role of AR in invasiveness is unknown. Almost 90% of cancer-related deaths are due to metastasis [2, 55]; thus, A better understanding of metastasis is essential for the development of targeted therapies to improve clinical outcomes.

Given the tremendous success of targeting estrogen in breast cancer, AR represents a possible therapeutic target in TNBCs and chemoresistant breast cancers, which have an otherwise inferior prognosis. However, the mechanism by which AR might contribute to TNBC metastasis is not understood and remains controversial. Moreover, the relationship between AR activation and actin reorganization via cofilin has not been studied nor is it clear if all TNBC cells from different origins would behave identically even if they expressed AR. In addition to using AR antagonists as single therapeutics, significant interest is being shown in potential strategies combining AR antagonists with other targeted treatments. Combination of phosphatidylinositol-3 kinase (PI3K)/mTOR inhibitors and AR antagonists exhibited synergetic activity in AR + TNBC and is currently under clinical investigation [10, 56], but a better understanding of how these pathways modulate actin dynamics that control metastasis is needed.

Since cofilin is a key player in the remodeling of actin filaments, ultimately driving the leading edge protrusion and invadopodia formation during cell escape from surrounding basal lamina, understanding if AR signaling impacts cofilin function in TNBC cell lines is critical to evaluating its possible use in therapy. We hypothesize that androgen receptor activation induces cell migration of AR+ TNBC cells through modulation of the amount, localization, and/or phospho-regulation of cofilin. Thus, this current work is focused on the role of activated AR during the migration and invasion of three AR+ TNBC cell lines and determining if altered cofilin amounts or phospho-regulation may explain quantifiable differences in migration behavior between these cell lines.

## Materials and methods

### Materials

All chemicals were reagent grade unless otherwise identified. Primary antibodies used were: mouse paxillin (1:250; BD Pharmingen, USA); mouse total cofilin (mAb22; 1:100–250) [57, 58]; affinity purified rabbit (4321) polyclonal phospho-cofilin (p-cofilin) (1:1000); mouse monoclonal AR (1:100; Santa Cruz); mouse monoclonal GAPDH (1:6000; CHEMICON, USA); affinity purified rabbit (1439) polyclonal that recognizes cofilin, p-cofilin, ADF and p-ADF with equal sensitivity (1:2000) [58]; rabbit polyclonal RhoA (1:1000; Proteintech); mouse monoclonal Rac1 (1:1000; Transduction Laboratories); rabbit monoclonal chronophin (1:1000; Cell Signaling); rabbit polyclonal phospho-LIM kinase (LIMK) 1/2 (1:1000, Cell Signaling); rat LIMK1 (1:500) [59]; rabbit polyclonal ROCK 1/2 (1:1000; Millipore); and rabbit slingshot-1L (SSH-1L R706; 1:500). Secondary antibodies used were: Alexa Fluor 488/594/647-conjugated goat anti-mouse/rabbit IgG (1:400; Invitrogen) for immunostaining and goat anti-mouse/rabbit Dylight 680/800 (1:10,000; Invitrogen); and HRP-conjugated goat anti-mouse/rat (1:10,000; Invitrogen) for western blotting.

### Cell culture

Three human AR+ TNBC cell lines: BT549, MDA-MB-453, and SUM159PT were obtained from University of Colorado Cancer Center (UCCC) Tissue Culture Core. Each cell line had

been propagated in a different medium making it impossible to directly compare their morphological and migratory responses to medium/substrate changes. Thus, all were adapted to grow on cell culture grade plastics and glass in high glucose Dulbecco's Modified Eagle Medium (HGDMEM, GIBCO, USA) supplemented with 10% fetal bovine serum (FBS; VWR, USA). Trypsin-EDTA (Sigma, USA) was routinely used for subcultures. Cell growth was accomplished at 37˚C in a 5% carbon dioxide/95% air atmosphere.

## Cell treatments

Cells were plated for 24 hr, washed twice with sterile phosphate buffered saline (PBS) and grown for 48 hr in 10% charcoal-stripped FBS (Sigma, USA)-containing HGDMEM. Cells were then treated for 72 hr unless otherwise noted. Cells were infected with adenovirus containing DNA oligonucleotide (5′–AAGTCTTCAACGCCAGAGGAG–3′) to generate short hairpin RNA (shRNA) for human cofilin-1 [60] at a multiplicity of infection (MOI) of 25. The adenovirus was constructed as described previously [61]. AR siRNA oligonucleotide (sc-29204, Santa Cruz) transfection was done using Lipofectamine RNAiMAX transfection reagent (Invitrogen), a non-targeting RNA (sc-44231, Santa Cruz) containing a scrambled sequence was used as a control. For androgen treatment, cells were treated with 10 nM or 100 nM DHT (Sigma, USA) or 10 nM of the AR agonist R1881 [(17b)-17-Hydroxy-17-methyl-Estra-4,9,11-trien-3-one] (Methyltrienolone, Metribolone, Sigma), with or without bicalutamide (10 µM, Santa Cruz) or adenovirus to silence cofilin. All pharmacologicals were prepared in dimethyl sulfoxide (DMSO), aliquoted, and stored at -20˚C. Experiments were performed within 2 months of reconstitution of androgens in DMSO. Final DMSO concentration in media did not exceed 0.1%.

## *In vitro* MTT cell proliferation assay

The MTT assay was used to evaluate cell proliferation [62]. Briefly, 1,000 cells/well were seeded in a 96-well tissue culture plate containing 10% FBS-HGDMEM and grown for 24 hr followed by 48 hr of growth in HGDMEM supplemented with 10% charcoal-stripped FBS. Cells were then treated for 24 hr, 48 hr, and 72 hr. Freshly prepared MTT salt (3-(4,5-dimethylthiazol-2yl)-2,5- diphenyltetrazolium bromide) (5 mg/ml; Sigma) was then added to each well to a final concentration of 0.5 µg/µL. The plates were incubated for 4 hr and the formation of formazan crystals was checked using an inverted microscope. An equal volume of 1:1 (200 µL) DMSO:isopropanol was added to each well and incubated for 30–45 min. From the absorbance of each well at 570 nm (BioTek EL309 MicroPlate Reader) the percent growth relative to untreated cells was calculated from the ratio of absorbance of the treated cells/vehicle treated control cells x 100. The experiment was performed three times in triplicates.

## Immunolabeling

Cells were plated on ethanol-flamed sterile glass cover slips for 24 hr and treated for 72 hr as previously described. Cells were fixed with 4% paraformaldehyde (Tousimis Research Corporation, USA) in PBS for 45 min, washed three times five min each with 0.1% Triton X-100 in PBS, fully permeabilized in 0.5% Triton X-100 in PBS for 10 min, washed with PBS and then blocked for 1 hr in 2% goat serum in PBS containing 1% bovine serum albumin (BSA). F-actin was stained with fluorescent-conjugated phalloidin (Invitrogen) in PBS for 1 hr. All antibodies were diluted in PBS containing 1% BSA. To visualize cell adhesions, fixed cells were incubated at room temperature for 1 hr with mouse anti-paxillin antibody, washed with PBS and incubated for 1 hr with fluorescent-conjugated goat anti-mouse IgG. For ratio imaging of p-cofilin/total cofilin, cells were incubated overnight at 4˚C with mouse total cofilin antibody

(mAb22) and rabbit 4321 polyclonal p-cofilin antibody. Microtubule immunostaining was achieved using mouse monoclonal anti-β-tubulin antibody (Sigma). Cells were then mounted with ProLong Gold Antifade containing DAPI (Invitrogen). Images were captured using 60× NA 1.4 objective on an inverted Nikon microscope with a CCD camera and operated by Metamorph software (Molecular Devices).

### Phospho-cofilin (p-cofilin)/total cofilin ratio image analysis

To assess p-cofilin/total cofilin ratio, the background of fluorescence was subtracted from each image, and the integrated intensity of the entire cells was normalized to equal 1 in ImageJ. A region of interest (ROI) from the cell edge to the rear of the leading lamellipodium (~ 100–150 nm) was drawn on the total-cofilin stained image, and the same ROI was used for p-cofilin stained image. Total integrated fluorescence intensities were averaged in that region. Three independent experiments with 15–20 migratory cells free of cell-cell contact in each experiment were used for analysis.

### Focal adhesion measurement

The size and number of focal adhesions per unit area (90 μm$^2$) were measured in Metamorph, an average of 10 unit areas at the leading edge of each cell were measured. Focal adhesion total area (μm$^2$)/90 μm$^2$ was measured as: average number of focal adhesions/μm$^2$ X average size of focal adhesions. The number of cells analyzed per experiment was ≥25 cells, three independent experiments.

### Adhesion assay

Seventy-two hours after treatment, cells were suspended in HGDMEM containing 0.35% BSA and replated onto 10 μg/mL collagen I (Sigma)-precoated 96-well culture dishes at a concentration of $5 \times 10^4$ cells/well. After incubation for 1 hr at 37˚C, dishes were washed twice with PBS, adherent cells were fixed in 4% paraformaldehyde for 30 min and stained with 1% methylene blue in 1% borax. After solubilization in 300 μL of 1% SDS, absorbance (630 nm) of the extract was measured [63, 64].

### Quantitative reverse transcription-polymerase chain reaction (qRT-PCR)

The quantity of *E-cadherin* and *N-cadherin* mRNA was determined by qRT-PCR [65]. Total RNA from control and treated cells was extracted using Direct-Zol RNA MicroPrep (Zymo Research) according to the manufacturer's instructions. After RNA extraction, cDNA was prepared using Power cDNA synthesis kit (iScript Reverse Transcription Supermix, Bio-Rad, USA). Amplification of target cDNA for epithelial-mesenchymal transition markers (*E-* and *N-cadherin*) and *GAPDH* (as a normalization gene) was done using iQ SYBR Green Supermix (Bio-Rad, USA). cDNA template (100 ng) was mixed with 150 nM of forward primer, 150 nM reverse primer, nuclease-free water and 10 μL master mix. PCR amplification was performed on CFX96 real-time PCR (Bio-Rad, USA) using the following thermal conditions: 3 min at 95˚ C and 40 cycles of 15 sec at 95˚ C, 30 sec at 58˚ C, and extension at 72˚ C for 30 sec, with a final extension at 72˚ C for 10 min for all primers. The experiment was performed three times in triplicates.

### Western blots

Control and treated cells were lysed in cold lysis buffer (2% SDS, 10 mM Tris pH 7.5, 10 mM NaF, 2 mM EGTA, 10 mM dithiothreitol). Cell extracts were heated in a boiling water bath for

5 min and sonicated. Aliquots of lysates were diluted in 4× SDS-PAGE sample buffer (0.5 M Tris–HCl pH 6.8, 2% SDS, 20% glycerol, 20% 2-mercaptoethanol and 0.16% bromophenol blue) and proteins resolved by electrophoresis on 10% or 15% SDS-polyacrylamide gels. Proteins were transferred onto nitrocellulose membranes and membranes blocked using 1% (w/v) BSA in Tris-buffered saline (TBS), and then incubated overnight at 4° C in primary antibodies diluted in TBS containing 1% BSA and 0.05% Tween 20 (TBST). After washing and incubation with appropriate IR-fluor-labeled secondary antibodies, immunolabeled bands were imaged using the LiCor Odyssey Infrared Imaging System. Signals were quantified using ImageJ software. For two dimensional western blots, proteins in SDS-containing cell extracts were precipitated with chloroform/methanol [66] and the protein pellet was rehydrated in 8 M urea, 2% IGEPAL, 18 mM dithiothreitol. Proteins were separated on a precast pH 3–10 focusing strips (Amersham Pharmacia Biotech, USA) according to the manufacturer's protocol (IPGphor Isoelectric Focusing System), followed by SDS-PAGE on 15% isocratic gels. After transfer onto nitrocellulose membrane and blocking, ADF and cofilin proteins were visualized with the rabbit 1439 pan ADF/cofilin antibody and IR-labeled secondary antibodies with the LiCor Odyssey.

## Invadopodia assay

Ethanol-flamed sterile 18 mm glass coverslips were placed in the wells of a 12-well tissue-culture plate and were coated with 50 μg/mL poly-L-lysine for 20 min at room temperature. The coverslips were then covered with 0.5% glutaraldehyde for 15 min, and then were coated with preheated (37˚C) 8:1 mixture of 0.2% porcine skin gelatin (Sigma, USA) with either 488- or 568-conjugated gelatin for 10 min at room temperature. The residual reactive aldehyde groups in the gelatin matrix were quenched with 5 mg/mL sodium borohydride in PBS for 15 min at room temperature [67]. Control and treated cells ($2 \times 10^4$ per coverslip) were incubated at 37˚ C for 6 hr (BT549 and SUM159PT cells) or 12 hr (MDA-MB-453 cells). Cells were stained for F-actin with fluorescent phalloidin labeled opposite to that of the fluor conjugated to the gelatin. Images were captured using 60× NA 1.4 objective on an inverted Nikon microscope with a CCD camera and operated by Metamorph software (Molecular Devices). The degradation area of gelatin ($\mu m^2$) was quantified in Metamorph and divided by number of cells in the same field and expressed as degradation area ($\mu m^2$)/cell. n $\geq$ 25 cells in each experiment, three independent experiments.

## Wound healing (scratch) assay

Control and treated cells were grown on a 6- well tissue culture plate to about 80–90% confluency. The monolayer was wounded by scraping with a 10 μL sterile plastic pipette tip. To avoid cell proliferation filling the wound, cells were treated with cytosine arabinoside (50 μM, Sigma) for 1 hr before the assay. Different fields were analyzed with Olympus CKX53 inverted microscope equipped with 4×, 0.13 NA air objective lens. Phase contrast images were captured using DMK 41BU02 camera (The Imaging Source) and acquired using Micro-Manager 1.4.22 Software. The effects of the different treatments were assessed by measuring the distance migrated by cells into the wound gap after a fixed time (usually 10–12 hr). The experiment was performed three times in triplicates.

## Transwell migration assay

Control and treated cells ($3 \times 10^4$) were seeded into the upper compartment of a 12-well chemotaxis chamber (Corning, USA). Both the upper and lower compartments were filled with HGDMEM containing 0.35% BSA and were physically separated by a polycarbonate

membrane (8.0 μm pore size) precoated for 4 hr with 100 μg/mL type I collagen. Cytosine arabinoside (50 μM, Sigma) was added to the upper chamber while epidermal growth factor (EGF, 5 nM, Invitrogen) was added to the lower chamber. Cells were incubated for 24 hr (BT549 and SUM159PT) or 36 hr (MDA-MB-453) at 37°C in 5% $CO_2$ humidified conditions. Non-migrating cells from the membrane upper surface were removed using a sterile cotton swab and the membrane was fixed with 4% paraformaldehyde and stained with 1% methylene blue in 1% borax. Migrating cells on the lower surface of the membrane from at least 30 fields/ membrane were counted using Olympus CKX53 inverted microscope equipped with 20×, 0.40 NA air objective [64, 68].

## Chemotaxis migration on thin layer of collagen IV

Control and treated cells (3 x 10^6 cells/mL) were suspended in HGDMEM containing 0.35% BSA and seeded on type IV collagen-coated chemotaxis μ-slide (ibidi GmbH, Martinsried, Germany) for 5 hr before imaging. HGDMEM without serum was used as the chemoattractant-free medium, and 10% FBS/HGDMEM was used as the chemoattractant-containing medium. Time-lapse video microscopy was performed using a 10x phase contrast objective on Keyence inverted microscope (BZ-X710), with a time interval of 10 min for 16–20 hr. The slide was inserted into Tokai Hit stage incubation system with the base and heated lid set to 37°C and with 5% $CO_2$ flowing at a rate of 10 L/hr. The following experiments were performed: a negative/negative (10% FBS -/-) control with no chemoattractant in either reservoir, a positive/positive (10% FBS +/+) control with 10% FBS in both reservoirs, a positive/negative (10% FBS +/-) control with 10% FBS in the left reservoir, and a positive/negative treated (androgen, cofilin KD, or both) with 10% FBS in the left reservoir. Cell tracking was performed in MATLAB using the image processing software CellTracker (http://celltracker.website/references.html). Cell trajectories were then imported into the ImageJ software plugin "Chemotaxis and Migration Tool" and extrapolated to (x,y) = (0,0) at time 6 hr, where the x-axis is parallel to the chemoattractant gradient and the y-axis is perpendicular to the gradient. On average 30–60 cells were tracked per experiment, and each experiment was repeated three times. To quantitatively evaluate directed cell migration, several values were generated: the forward migration indices parallel (FMI^II) and perpendicular (FMI^) to the direction of the chemoattractant gradient, cell velocity, percentage of cells moving toward the chemoattractant, and the Rayleigh test [69]. The forward migration indices represent the efficiency of cells migrating relative to the direction of the chemoattractant gradient (ibidi.com/manual-image-analysis/171-chemotaxis-and-migration-tool.html). The FMI^II values were calculated by dividing the final distance from the origin in the direction parallel to the chemoattractant gradient by the total distance traveled for each cell, averaged over the total number of cells analyzed. The FMI^ values were calculated similarly using the final distance from the origin in the direction perpendicular to the chemoattractant gradient for each cell. The Rayleigh test was used to assess the uniformity of the circular distribution of the endpoints of the cell trajectories. An experiment was considered to exhibit statistically significant chemotaxis when the FMI^II value was larger in magnitude than the FMI^ value and the Rayleigh test p-value was less than 0.05.

## Statistical analysis

Results were expressed as mean ± SEM. Statistical analysis was performed using GraphPad Prism version 7.0 (GraphPad Software, San Diego, CA, USA). To determine differences between 3 or more means, one-way ANOVA with Fisher's LSD for multiple comparisons post-tests was performed. P value < 0.05 was considered significant.

## Results

### Expression of androgen receptor (AR) protein differs among TNBC cell lines

We examined the levels of AR protein in the three different TNBC cell lines (BT549, MDA-MB-453, and SUM159PT) and compared these to AR protein in the ER+/AR+ MCF7 breast cancer cell line by 1D western blots normalizing the AR band intensity to that of GAPDH as a loading control (Fig 1A). The amounts of AR protein expressed by BT549 and SUM159PT cell lines are not significantly different from MCF7 cells (Fig 1B). However, and as expected, the luminal androgen receptor (LAR) MDA-MB-453 cell line, contains the highest AR protein level, 3.2 fold more than MCF7 cells (Fig 1A and 1B) [12].

### TNBC cell lines express different levels of total, phospho- and dephospho-cofilin and treatment with androgens demonstrate cell-line specific effects

The levels of total cofilin protein (dephosphorylated and phospho-cofilin) in the different TNBC cell lines were examined by 1D western blots normalizing the cofilin band to that of GAPDH as a loading control (Fig 1C). MDA-MB-453 cells express 1.6-fold more cofilin compared to BT549 and SUM159PT cells (Fig 1D). Moreover, we examined the levels of dephosphorylated (active) cofilin and ADF (if present) and their phosphorylated forms in extracts of the three TNBC cell lines and in a rat mammary tumor cell line (MTLn3) which we previously characterized as expressing both ADF and cofilin by 2D western blots [64] (Fig 2A). We used a polyclonal antibody (1439) that recognizes ADF and cofilin (and their phosphorylated forms) with about equal sensitivity [58] as well as a cofilin-specific monoclonal antibody (mAb22) which does not recognize ADF. As shown in Fig 2A and 2B, ADF and cofilin-1 (hereafter referred to as cofilin) in MTLn3 cells are expressed nearly equally and are phosphorylated to a nearly equal extent. Each of the TNBC cell lines expresses cofilin but not ADF (Fig 2A). BT549 and MDA-MB-453 cells contain significantly more (1.5 to 3 fold) p-cofilin than dephosphorylated cofilin (Fig 2B), whereas SUM159PT cells contain nearly equal amounts of each form (Fig 2B). Treating TNBC cells with 10 nM DHT, with or without bicalutamide, did not significantly alter total cofilin levels as compared to untreated cells (Fig 1C and 1D) ($p \geq 0.05$). However, BT549 cells treated with DHT (10 and 100 nM) or R1881 showed a significant ($p < 0.05$) increase in the pool of dephosphorylated cofilin (Fig 2C) as judged from 2D western blots (S1 Fig). MDA-MB-453 cells showed a similar response to 10 nM DHT and R1881 but not to 100 nM DHT. Androgen treatment had no significant effect on the cofilin/p-cofilin amounts in SUM159PT cells (Fig 2C) ($p \geq 0.05$).

### Knock down of cofilin and AR expression in TNBC cells and effect of cofilin knock down on AR levels

To investigate the roles of AR and cofilin in the invasive phenotype of TNBC cells, we used adenoviral-mediated shRNA expression to silence cofilin and siRNA transfection to silence AR. Western blots of extracts from TNBC cells so treated (S2A and S2B Fig) indicate knock down (KD) of greater than 90% for cofilin and 80% for AR was achieved by 72 hr post-treatment (S2C Fig). Cofilin and AR levels in cells expressing a non-silencing RNA as a control were not significantly different at 3 days from the 0-time uninfected cells (S2C Fig), demonstrating that the viral infection or RNA transfection per se had no effect on cofilin or AR expression. To determine if silencing cofilin directly affected the expression of AR, we analyzed by western blots the level of AR in each of the TNBC cell lines expressing cofilin silencing shRNA (KD) (S2D and S2E Fig). Although there were slight differences in the AR expression in response to cofilin silencing, none of them reached a level of significance above $p < 0.05$.

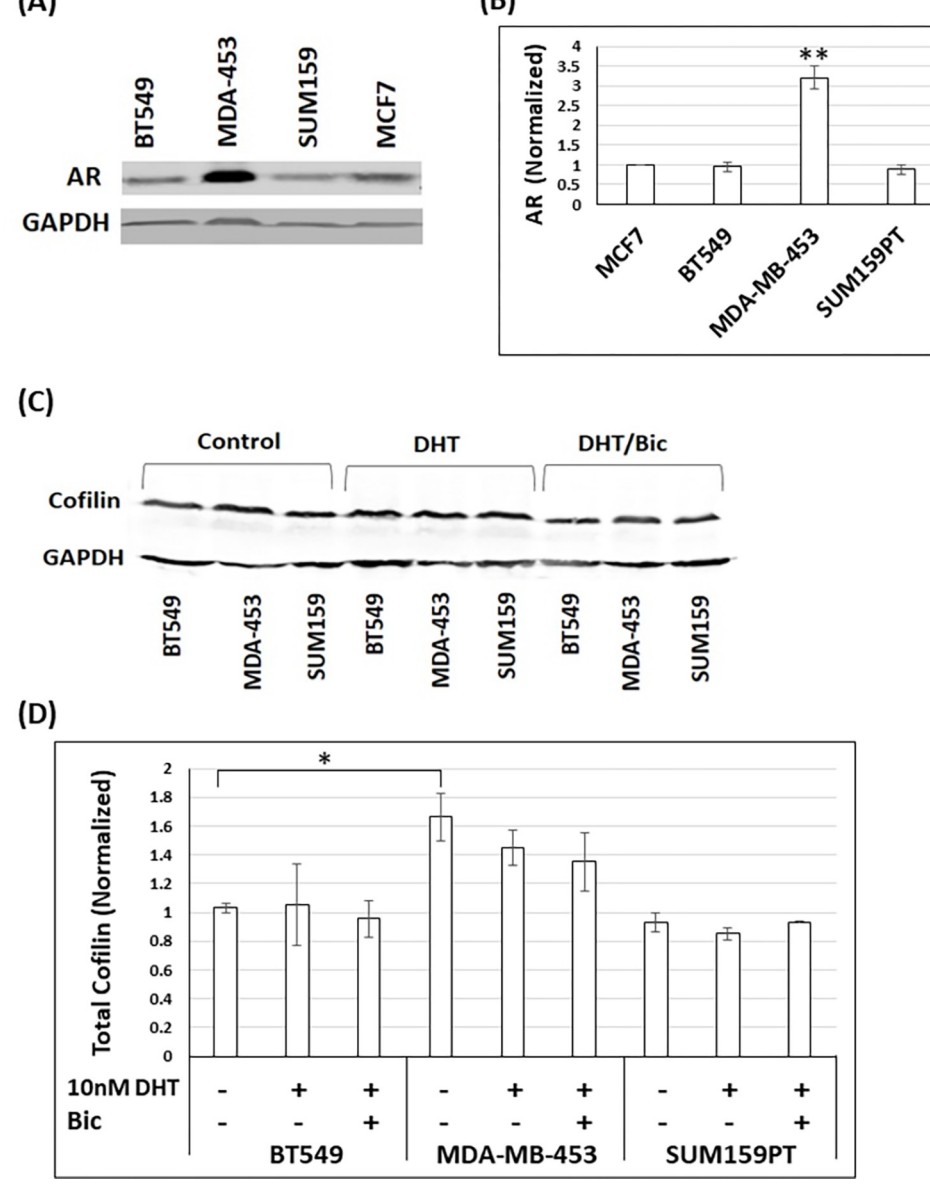

**Fig 1. TNBC cell lines express different protein levels of AR and cofilin-1 (cofilin).** (A) Representative western blot of lysates from vehicle-treated TNBC and MCF7 cells immunolabeled with AR antibody and a monoclonal antibody to GAPDH as a loading control. The experiment was repeated three times and the corresponding quantification is shown in (B). (B) Quantification of AR protein level in the different TNBC cell lines relative to its level in MCF7 cells [set as 1.0 arbitrary unit (a.u.)] obtained from densitometers of immunoblots. ** $p < 0.01$ versus MCF7 cells. (C) Representative western blot of lysates from control vehicle-treated, 10 nM DHT (DHT)-treated, and 10 nM DHT/ bicalutamide (Bic)-treated TNBC cells immunolabeled with mAb22 antibody and GAPDH as a loading control. The experiment was repeated three times and the corresponding quantification is shown in (D). (D) Quantification of total cofilin protein levels in TNBC cells relative to its level in control BT549 cells (set as 1.0 a.u.) obtained from densitometers of immunoblots. * $p < 0.05$ versus control BT cells.

## Androgen treatment modulates the upstream pathways of cofilin phospho-regulation differently among TNBC cell lines

Levels of Ser3 phospho-cofilin are regulated by a complex balance of kinase and phosphatase activity, either of which can be regulated by other upstream factors [36]. To determine if the

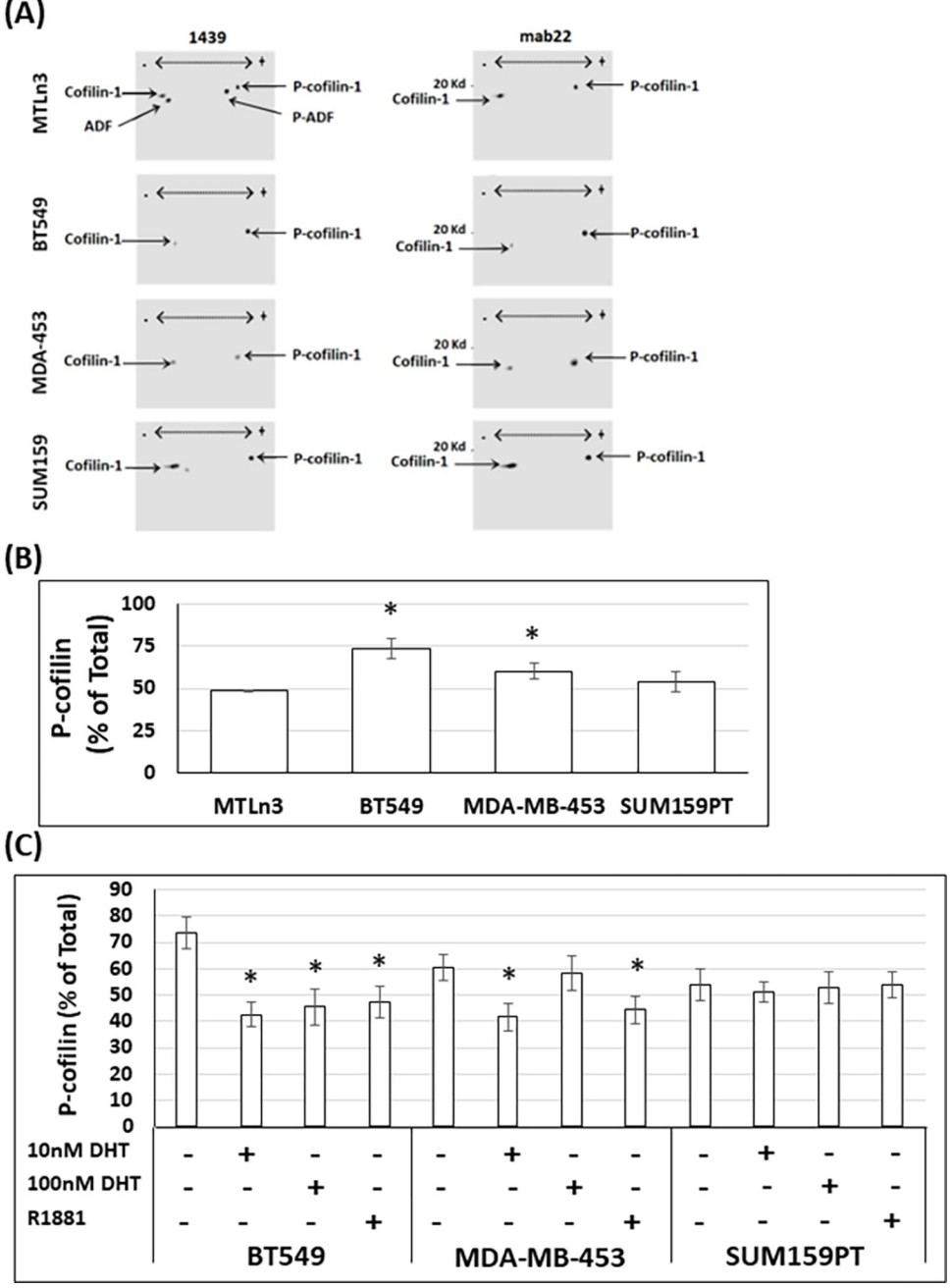

**Fig 2. TNBC cell lines express different protein levels of cofilin phosphorylated form and androgen treatment changes these protein levels.** (A) Representative 2D western blots of extracted TNBC and rat MTLn3 cell proteins immunolabeled with rabbit polyclonal antibody (1439) that recognizes ADF, phospho-ADF, cofilin, and phospho-cofilin with equal sensitivity or a monoclonal antibody to total cofilin (mAb22). The experiment was repeated three times and the corresponding quantification is shown in (B). (B) Quantification of cofilin level and its phosphorylated form in vehicle-treated control TNBC cells obtained from densitometers of immunoblots. * p < 0.05 versus MTLn3 cells. (C) Quantification of levels of cofilin and its phosphorylated form in androgen-treated cells as compared to vehicle-treated control TNBC cells obtained from densitometers of 2D immunoblots (S1 Fig). The experiment was repeated three times. * p <0.05 versus control.

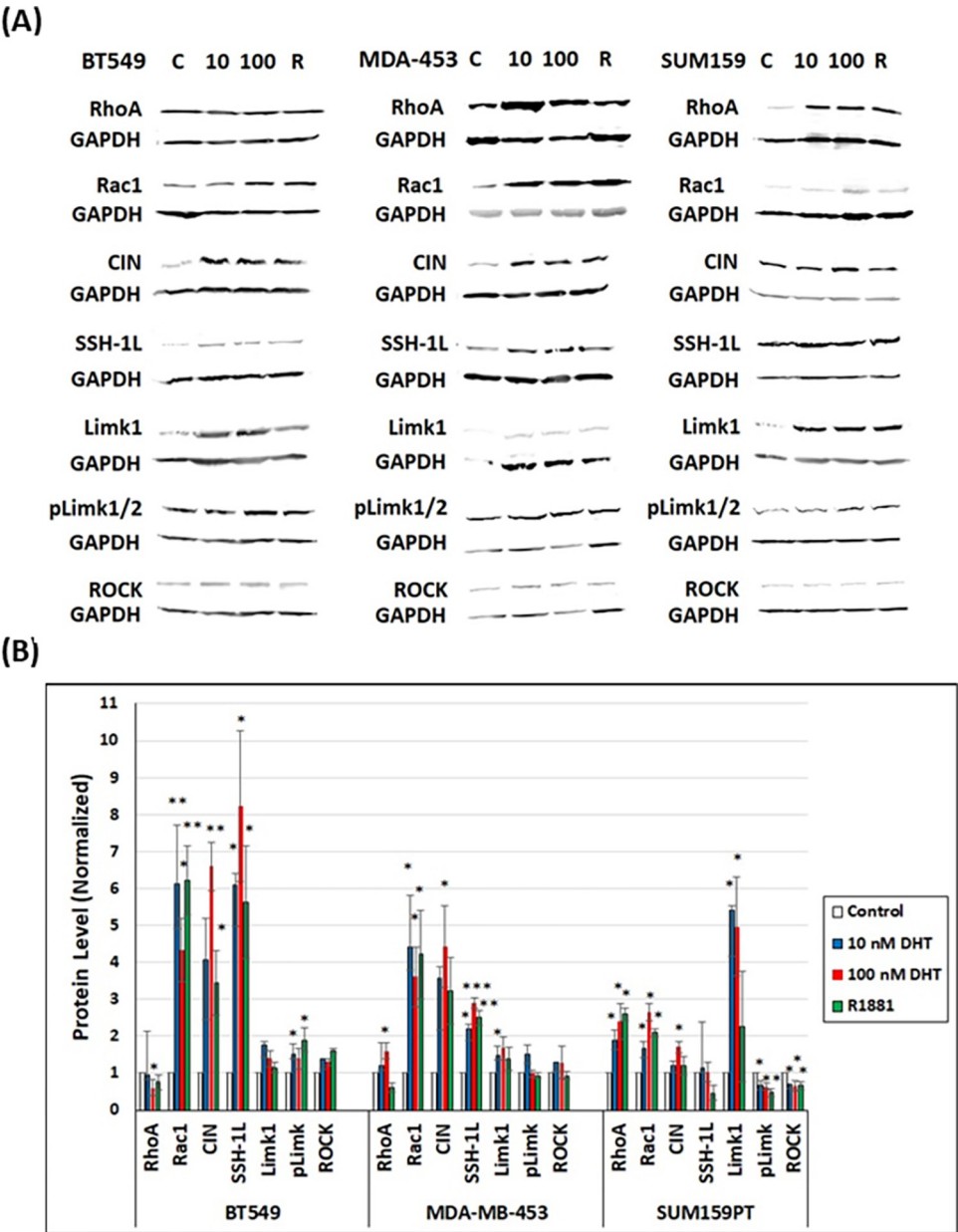

**Fig 3. Androgen treatment changes protein levels of cofilin phosphatases in BT549 and MDA-MB-453 and LIMK1 in SUM159PT cells.** (A) Representative western blots of lysates from vehicle- and androgen-treated TNBC cells immunolabeled with antibodies against different cofilin regulators. The experiment was repeated three times and the corresponding quantification is shown in (B). (B) Quantification of protein levels of different cofilin regulators (Total RhoA, total Rac1, chronophin (CIN), slingshot (SSH), total LIMK1 (LIMK1), pLIMK1/2, and ROCK) in androgen-treated cells as compared to control cells (set as 1.0 a.u.) obtained from densitometers of immunoblots. * p <0.05, ** p < 0.01, *** p <0.001 versus control.

different TNBC cell lines utilize the same or different pathways for cofilin phospho-regulation, we examined the protein levels, relative to GAPDH, of cofilin upstream effectors: total RhoA, total Rac1, CIN, SSH-1L, total LIMK1, active p-LIMK1/2 and ROCK 1/2 by western blotting of extracts from control and androgen-treated TNBC cells (Fig 3A). In BT549 cells, androgens changed the ratio of cofilin:p-cofilin 3.6 fold (from 1:3 to 1.2:1) (Fig 2C averaged over the two

DHT concentrations and R1881) and this is accompanied by an increase in the protein levels of two major cofilin phosphatases chronophin (CIN) and slingshot (SSH-1L) as well as the upstream regulator Rac1 (Fig 3B). A slight but significant increase of pLIMK (active) and ROCK was also quantified in androgen treated BT549 cells, but this <2-fold increase was counteracted by the 4-to-6-fold increase in cofilin phosphatases (Fig 3B). A similar trend but with smaller increase in total Rac1, CIN and SSH-1L levels occurred in androgen-treated MDA-MB-453 cells which may account for the smaller but significant androgen-induced increase in active dephosphorylated cofilin in this cell line (Fig 2C). In the SUM159PT cells, total RhoA and LIMK1, which were only modestly altered by androgens in the other two lines, showed the greatest androgen-induced increase. Surprisingly, the increase in total LIMK1 was not reflected by an increase in the T508 phosphorylated form required for its activity. Indeed, a slight but significant decrease was found in pLIMK1 as well as in ROCK (Fig 3B), suggesting the increase in total RhoA protein was not accompanied by its activation to the GTP form required for ROCK activation [70]. The slight changes in the levels of cofilin kinase/phosphatases in SUM159PT cells were apparently insufficient to alter the cofilin:p-cofilin ratios, which stayed constant at about 1:1 following androgen treatment (Fig 2C and S1 Fig).

## Effects of androgens on the intensity ratio of phospho-cofilin/total cofilin at the leading edge of TNBC cells

Next, we investigated if androgens locally regulate cofilin activity at the leading edge of migratory TNBC cells. We examined the localization of p-cofilin (inactive; Fig 4A and 4E) and total cofilin (Fig 4B and 4F) in control and androgen-treated TNBC cells. Quantification of the fluorescence intensity ratio of p-cofilin to total cofilin (Fig 4C and 4G) at the leading edge (within about 200 nm of the membrane) [Fig 4D and 4H, ROI (Region of Interest)] showed an increase of p-cofilin/total cofilin ratio in androgen-treated cells by 2.5- and 5.6-fold in BT549 and MDA-MB-453 cells, respectively (Fig 4I averaged over the two DHT concentrations and R1881) compared to control cells. However, this increase in p-cofilin/total cofilin was not observed in SUM159PT cells (Fig 4I).

Based on previous preliminary observations indicating that microtubules contribute to localization of p-cofilin at the leading edge, p-cofilin and microtubules were immunolabeled and their localization was compared between polarized and non-polarized (apolar) cells in BT549 and SUM159PT cells (Fig 5). At the leading edge of apolar BT549 cells (Fig 5A–5D), microtubules do not maintain a perpendicular orientation to the membrane and often run in parallel bundles along the membrane (Fig 5D, white arrows). Although p-cofilin appears to be enriched near the membrane of apolar BT549 cells (Fig 5A), it is more uniform around the entire lamellipodium, whereas in the polarized BT549 cells, the regions of enriched p-cofilin (Fig 4E) are associated with the furthest protruding membrane (arrows in Fig 5H). In polarized cells, p-cofilin is localized beyond microtubule tips (Fig 5H) suggesting the possibility that microtubules delivery of cofilin activating proteins helps establish the dynamics of actin turnover and membrane protrusion. Furthermore, tips of microtubules are stabilized proximal to focal adhesions behind the leading-edge protrusions and are essential for focal adhesion dynamics modulating cell polarity [71]. In contrast, a clear distinction between p-cofilin-rich area at the cell periphery versus microtubule tips was not observed in SUM159PT cells (Fig 5I–5P).

## Effects of cofilin knock down on androgen-mediated changes in TNBC cell proliferation

Since cofilin is essential for eukaryotic cell division [72–76] and androgens have been shown to enhance cell proliferation in TNBC [77–79], we analyzed the effects of knocking down

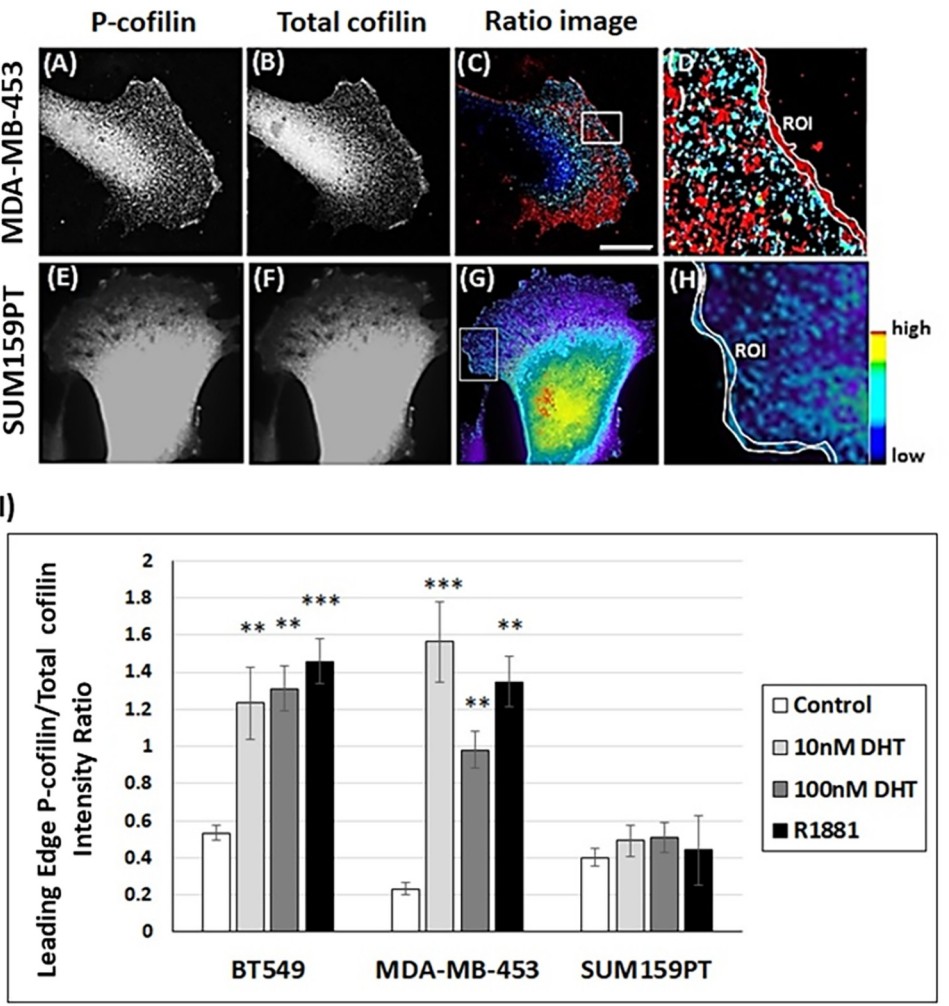

**Fig 4. Androgen treatment increases p-cofilin/total cofilin ratio at the leading edge of BT549 and MDA-MB-453 cells.** Representative images showing localization of p-cofilin in polarized (A) MDA-MB-453 cells and (E) SUM159PT cells, and total cofilin in MDA-MB-453 cells (B) and (F) SUM159PT cells. (C, G) P-cofilin/total cofilin ratio images from which the fluorescence intensities at the leading edge [ROI (Region of Interest) in (D, H)] were measured, scale bar: 5 μm. (D, H) Enlarged ratio image of the white box in (C and G, respectively) where ROI from the cell edge to the rear edge of the leading edge (150–200 nm) was drawn. (I) Calculated fluorescence intensity ratios of p-cofilin/total cofilin at the leading edge of migratory BT549, MDA-MB-453 and SUM159PT cells free of cell-cell contact. An average of 15–20 cells were analyzed, three independent experiments. ** p <0.01, *** p <0.001 versus control.

cofilin on TNBC cell proliferation (S3 Fig) and multinucleation (S4 Fig) and if there is a cell-line specific response to androgens.

Cells were either infected with adenoviruses expressing cofilin shRNA, or treated with the different androgens, or both at day 0 and then cell proliferation was followed at 24 hr intervals for 3 days (S3 Fig); the percentage of uni- and multinucleated cells was scored after 72 hr of treatment (S4 Fig). Both BT549 and MDA-MB-453 cell proliferation increased about 1.5 fold in response to 10–100 nM DHT and R1881 in the first 24 hr (S3 Fig). In contrast, SUM159PT cells showed a more modest (1.2 fold) but significant increase in growth rate to all androgens (S3 Fig) during the first 24 hr but increased by day 2 and remained in the 1.2 to 1.5 fold increase range, similar to the other lines, on day 3 (S3 Fig). Bicalutamide inhibited the andro-gen-induced increase in growth rate and thus it appeared to be AR mediated (S3 Fig). By three

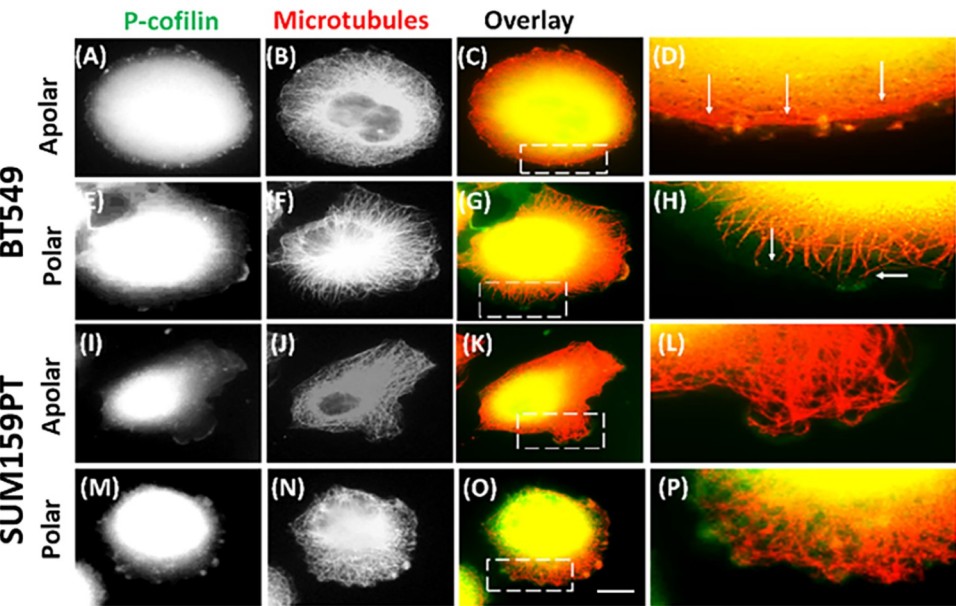

**Fig 5. Microtubules contribute to localization of p-cofilin at the leading edge of BT549 and MDA-MB-453 cells.**
Representative images showing localization of p-cofilin in apolar BT549 cells (A), polar BT549 cells (E), apolar
SUM159PT cells (I), polar SUM159PT cells (M), microtubules (B, F, J, N, respectively), and overlay (C, G, K, O,
respectively). Images D, H, L, P are enlarged images of the dashed white boxes in C, G, K, O, respectively. White
arrows in images D and H indicate orientation of microtubules at the leading edge. MDA-MB-453 cells (not shown)
have similar distributions to BT549 cells. Scale bar: 10 μm.

days following expression of cofilin shRNA, all three TNBC cell lines grew significantly slower
than the controls in the absence or presence of androgens (S3 Fig) indicating that activated AR
and cofilin affect cell division independently. Multinucleated cells were about twice as preva-
lent in the MDA-MB-453 line (27%) compared to BT549 (12.3%) and SUM159PT (16%) cells
and were decreased in DHT-treated MDA-MB-453 and SUM159PT lines but not in the BT549
cells (S4 Fig). Surprisingly, R1881 reduced multinucleation in BT549 and SUM159PT cells but
not in MDA-MB-453 cells. These androgen mediated effects were mostly eliminated with bica-
lutamide co-treatment as well as by silencing androgen receptor, confirming their dependency
on AR (S4 Fig). As expected from cofilin's role in cytokinesis, cofilin KD in each cell line
showed a time-dependent increased multinucleation (S4 Fig) which was not inhibited by
androgen treatment.

### Effects of cofilin knock down on androgen-induced morphological changes

Changes in cell shape associated with tumor metastasis include development of a polarized
phenotype with a leading edge and cell rear and an accompanied cell elongation, all driven by
cytoskeletal reorganization involving both microtubule/focal adhesion dynamics [71] and the
actin filament network [80, 81]. To elucidate the effects of cofilin KD, androgens, and their
combination on cell shape, actin cytoskeletal organization, and cell polarization, cells were
treated and stained for actin filaments using fluorescent phalloidin.

BT549 cells grow as individual cells that show some shape heterogeneity with the major
form being cuboidal epithelium exhibiting cortical actin cytoskeleton (S5A Fig). Polarized
cells, crescent (S5B Fig) or kite-like (S5C Fig) have thin lamellipodia, and the majority (73.3%)
of polarized BT549 cells had the kite-like morphology (S5Q Fig). MDA-MB-453 cells grow as
clusters (grape-like or stellate structures, S5G Fig) but individual cells show prominent cortical

actin (S5H Fig); moreover, more than 80% of polarized MDA-MB-453 cells exhibit the crescent morphology (S5I Fig) and are less kite-like (S5J and S5Q Fig). SUM159PT cells grow mainly in groups (S5L Fig), but individual cells do not have major regions of phalloidin-stained F-actin except directly along the cell periphery (S5M Fig). Around two thirds (68.5%) of polarized SUM159PT cells exhibit the kite-like morphology (S5O Fig) rather than the crescent morphology (S5N and S5Q Fig). Knocking down cofilin in the three TNBC cell lines resulted in the formation of thick actin stress fibers and cell rounding (S5D, S5K and S5P Fig). On the other hand, androgen treatment caused the cells to elongate (S5Q Fig) and exhibit kite-like morphology (S5E, S5J and S5O Fig). Interestingly, many androgen-treated BT549 cells underwent a dramatic change in cell morphology with an elongated cytoplasmic extension that resembled a neurite (S5E Fig).

Cells were classified as apolar (non-polar): having either no lamellipodium or more than two lamellipodia (S5A, S5D, S5H, S5K, S5M and S5P Fig); bipolar: having two lamellipodia (S5F Fig); or polar: having one lamellipodium generating one leading edge (S5B, S5C, S5E, S5I, S5J, S5N and S5O Fig). Androgen treatment of TNBC cells significantly increased the polar phenotype while decreasing the % of apolar cells with little effect on bipolar phenotype (Fig 6A–6C). The reverse was noticed in cells where cofilin expression was silenced (cofilin KD). The effect of androgens on cofilin KD-reduced polarization was cell line-specific. In MDA-MB-453 cells with cofilin KD, androgen treatment increased the percentage of polar cells to control levels, had no effect in SUM159PT cells, and failed to restore the % of polarization to that found in control BT549 cells (Fig 6A–6C). SUM159PT cells showed greater than 75% polarized phenotype following 100 nM DHT treatment in both control and cofilin KD cells, which along with the other evidence presented, suggests that these cells use a cofilin-independent polarization mechanism. The androgen effect on polarization of the three TNBC cell lines was reduced to near control levels with AR silencing or bicalutamide treatment except in the case of the highest does of DHT (Fig 6A–6C).

Additionally, cell area, and the ratio of length/breadth were measured as these are also important indicators of cytoskeletal organization (S6 Fig). Treating BT549 or MDA-MB-453 cells with androgens had no effect on cell area, but all androgens significantly increased the cell length/breadth ratio, an effect largely abrogated by bicalutamide (except in the case of the highest dose of DHT) or silencing of AR (S6A–S6D Fig). BT549 and MDA-MB-453 cells with cofilin KD, with or without androgen treatment, had significantly increased areas but no significant changes in length/breadth ratio (S6A–S6D Fig). Androgen-treated SUM159PT cells and cofilin KD cells with or without androgens showed increased cell area but had no effect on length/breadth ratio from untreated cells (S6E and S6F Fig).

## Effects of cofilin knock down on androgen-mediated changes in cell adhesion

Cell adhesion is an essential step during cell migration [82], thus we studied the effects of androgen treatment with or without cofilin KD on TNBC cell adhesion. Cells were stained with anti-paxillin antibody (Fig 7B and 7E), and the size and number of focal adhesions were measured in the same total area at the leading edge of similarly shaped cells (Fig 7M, 7O and 7Q). In addition, control and treated cells were seeded onto collagen I coated dishes, and adherent cells were quantified after 1 hr (adhesion assay) and normalized to control set at 100% (Fig 7N, 7P and 7R).

BT549 and MDA-MB-453 cells treated with the different androgens showed decreased focal adhesion area compared to untreated (control) cells and the response was fully reversed by AR silencing and by bicalutamide except in the cells treated with highest (100 nM) dose of

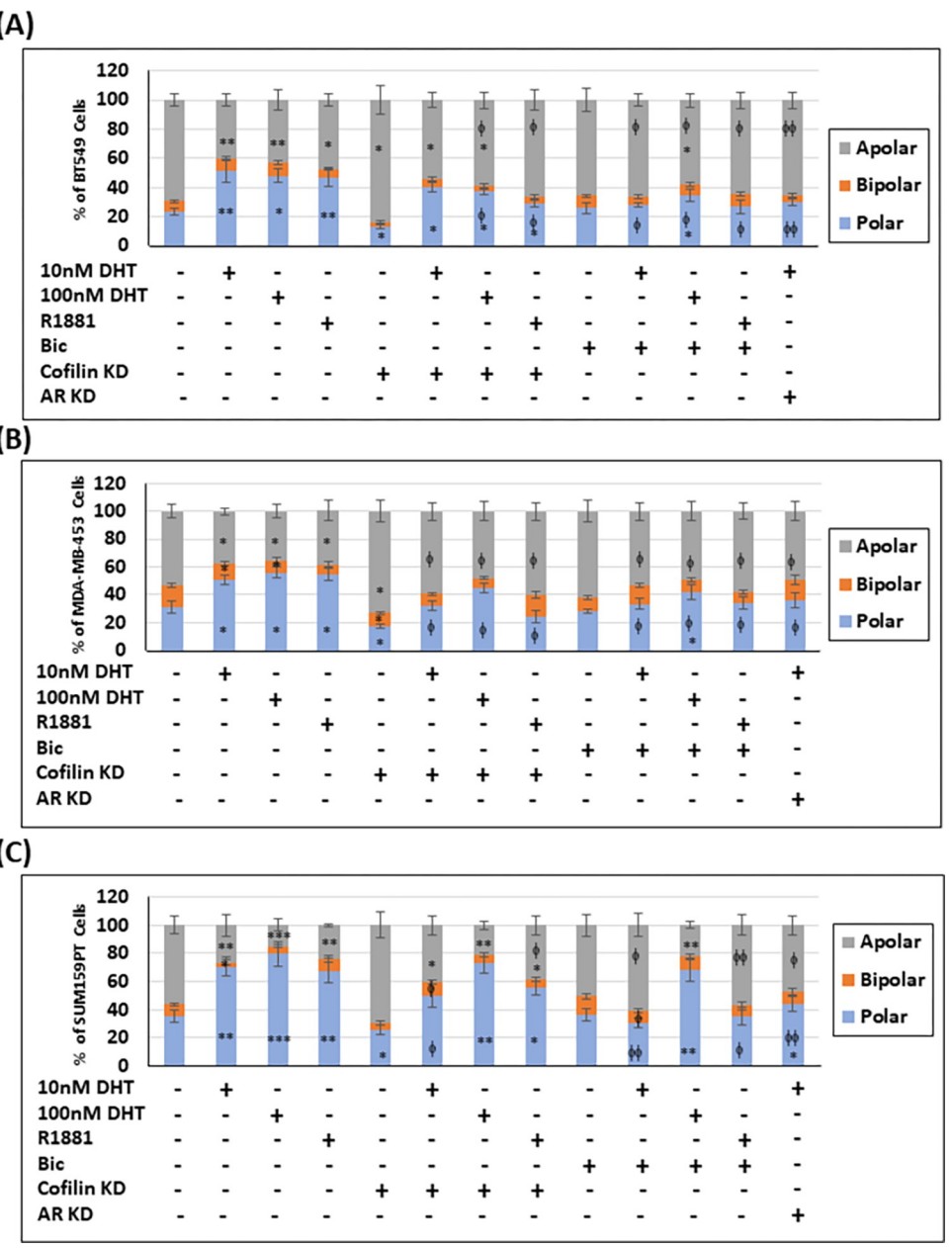

**Fig 6. Knocking down cofilin in androgen-treated MDA-MB-453 cells restored the control polarization phenotype but not in BT549 or SUM159PT cells.** Control and treated TNBC cells were stained with fluorescent-conjugated phalloidin and categorized as apolar, bipolar or polar (see S5 Fig), and the percentage of cells in each category was scored in BT549 cells (A), MDA-MB-543 cells (B), and SUM159PT cells (C) and presented as percentage of total cells. Data are expressed as mean ± SEM, n ≥ 100 cells in each experiment, three independent experiments. * $p$ <0.05, ** $p$ <0.01, *** $p$ <0.001 versus control, $^{\phi}$ $p$ <0.05, $^{\phi\phi}$ $p$ <0.01 versus androgen-treated cells.

DHT (Fig 7M and 7O and S7 Fig). SUM159PT cells did not show any significant effect of androgens on focal adhesion area, nor did inclusion of bicalutamide or silencing AR significantly alter adhesions in SUM159PT cells (Fig 7Q). However, all three cell lines showed increased focal adhesion areas upon cofilin KD. Androgens fully reversed the effects of silencing cofilin expression in BT549 cells, but not in MDA-MB-453 and SUM159PT cell lines (Fig 7M, 7O and 7Q and S7 Fig), again suggesting the AR-dependent differences in signaling

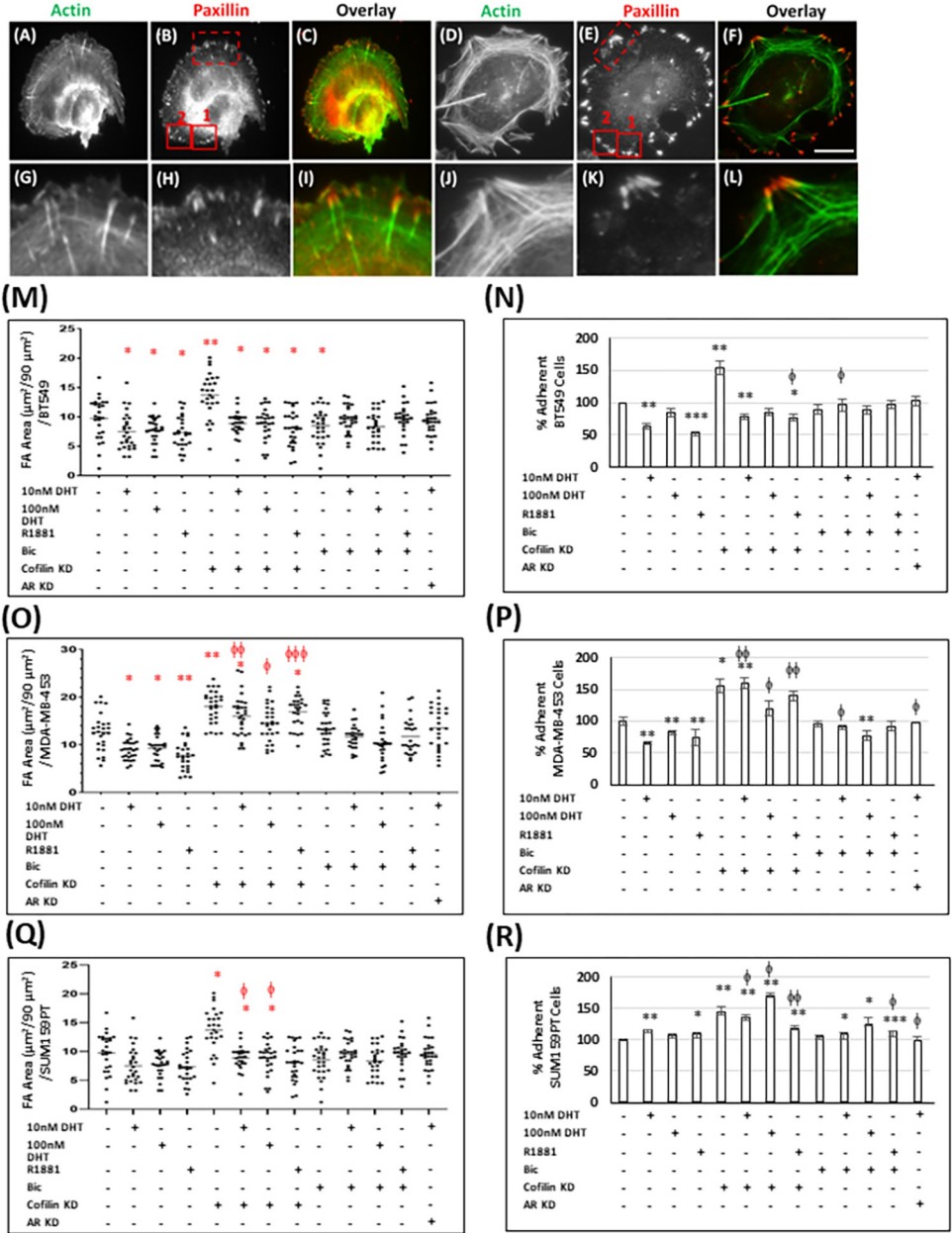

**Fig 7. Cofilin suppression enhances TNBC cell adhesion, while androgen treatment of BT549 and MDA-MB-453 cells decreases it.** (A-C) Representative images of regular focal adhesion in TNBC cells. (D-F) Representative images of enhanced focal adhesion in TNBC cells. TNBC cells were stained for actin with fluorescent-conjugated phalloidin (A, D), and for focal adhesion with mouse anti-paxillin primary antibody (B, E) (red dashed box, enlarged in G-L). The size and number of focal adhesions per unit area (90 $\mu m^2$; red solid boxes in B and E) were measured in Metamorph. Scale bar: 10 $\mu$m. (M, O, Q) Dot plots showing the quantification of total area occupied by focal adhesion ($\mu m^2$)/ 90 $\mu m^2$ in BT594 (M); MDA-MB-453 (O), SUM159PT (Q), n $\geq$ 25 cells. Collagen I adhesion assay was performed for BT549 (N), MDA-MB-453 (P), SUM159PT (R). Average values of cells adherent to plastic (not shown) were subtracted from average values adherent to collagen I. Average values of control cells were reported to 100%. Three independent experiments each performed in triplicate. * p <0.05, ** p < 0.01, *** p <0.001 versus control, $^\phi$ p <0.05, $^{\phi\phi}$ p <0.01, $^{\phi\phi\phi}$ p <0.001 versus androgen-treated cells.

pathways (in this case, a cofilin-mediated pathway) among these lines probably extend to regulators of focal contacts and maturation of focal adhesions. Since microtubules appear to have a role in dynamics of focal adhesions through plus-TIP delivery of essential factors [71], the different orientations of microtubules between the BT549 (Fig 5A–5H) and SUM159PT cells (Fig 5I–5P) might contribute significantly to the differences in adhesion area.

Results from the cell adhesion assay (Fig 7N, 7P and 7R), especially if looked at by the patterns of the results, were quite similar overall to those measuring focal adhesion area. BT549 and MDA-MB-453 cells again showed the most pronounced inhibition (reduced adhesion) by androgens with recovery to at or near control levels with AR silencing or bicalutamide treatment except at the highest DHT dose (Fig 7N and 7P). The enhanced cell adhesion observed in cofilin KD BT549 cells was abrogated by androgen treatment, but this was not found in MDA-MB-453 or SUM159PT cofilin KD cells (Fig 7N, 7P and 7R and S7 Fig).

## Effect of expressing cofilin shRNA on androgen-enhanced *N-cadherin/E-cadherin* mRNA ratio

During breast cancer metastasis, cancer cells undergo epithelial-mesenchymal transition (EMT) [83] during which there is a downregulation of E-cadherin and up-regulation of N-cadherin, i.e., cadherin switching [84]. Here we examined the effects of androgens and cofilin KD on *N-cadherin/E-cadherin* mRNA (*N/E*) ratio by RT-PCR (Fig 8). BT549 and MDA-MB-453 cell lines showed a15-25 fold increase in the *N/E* ratio in response to DHT (10–100 nM) and a smaller but significant increase (4–7 fold) in response to R1881. These responses were shown to be AR dependent by both AR silencing and bicalutamide treatment (Fig 8A and 8B). SUM159PT cells showed no androgen response (Fig 8C). Cofilin KD in both BT549 and MDA-MB-453 cells abrogated the androgen response, while in SUM159PT cells cofilin KD reduced the *N/E* ratio to well below the control whether androgens were present or not (Fig 8).

## Androgen treatment does not change matrix degradation activity in TNBC cells

Invasive cancer cells form actin-rich membrane protrusions with matrix degradation activity known as invadopodia (S8A Fig). Since cofilin is needed for invadopodia initiation, stabilization, and maturation [85, 86], and silencing cofilin expression was found to interfere with long-lived invadopodia in metastatic carcinoma cells [64, 87], we determined if androgen treatment of TNBC cells silenced for cofilin affect invadopodia formation and matrix degradation activity.

In BT549 and MDA-MB-453 cells there was no significant effect of androgens on the degradation area of collagen and SUM159PT cells only showed a response to the highest concentration of DHT (S8B–S8D Fig). As expected, cofilin KD significantly decreased the degradation area in all three lines with no effects of androgens in BT549 cells, a slight increase with R1881-treatment in MDA-MB-453 cells and a slight further decrease in degradation area in androgen treated SUM159PT cells. To confirm the lack of involvement of androgens in invadopodia information, neither AR silencing nor bicalutamide treatment had any effect (S8 Fig).

## Effects of knocking down cofilin on androgen-mediated changes of TNBC cell migration

To understand how cofilin KD with or without androgen treatment affects migration of each TNBC cell line, 3 different migration assays, wound healing assay (Fig 9 and S9 Fig) as an example of collective migration, transwell migration (Fig 9 and S9 Fig) and chemotaxis μ-slide

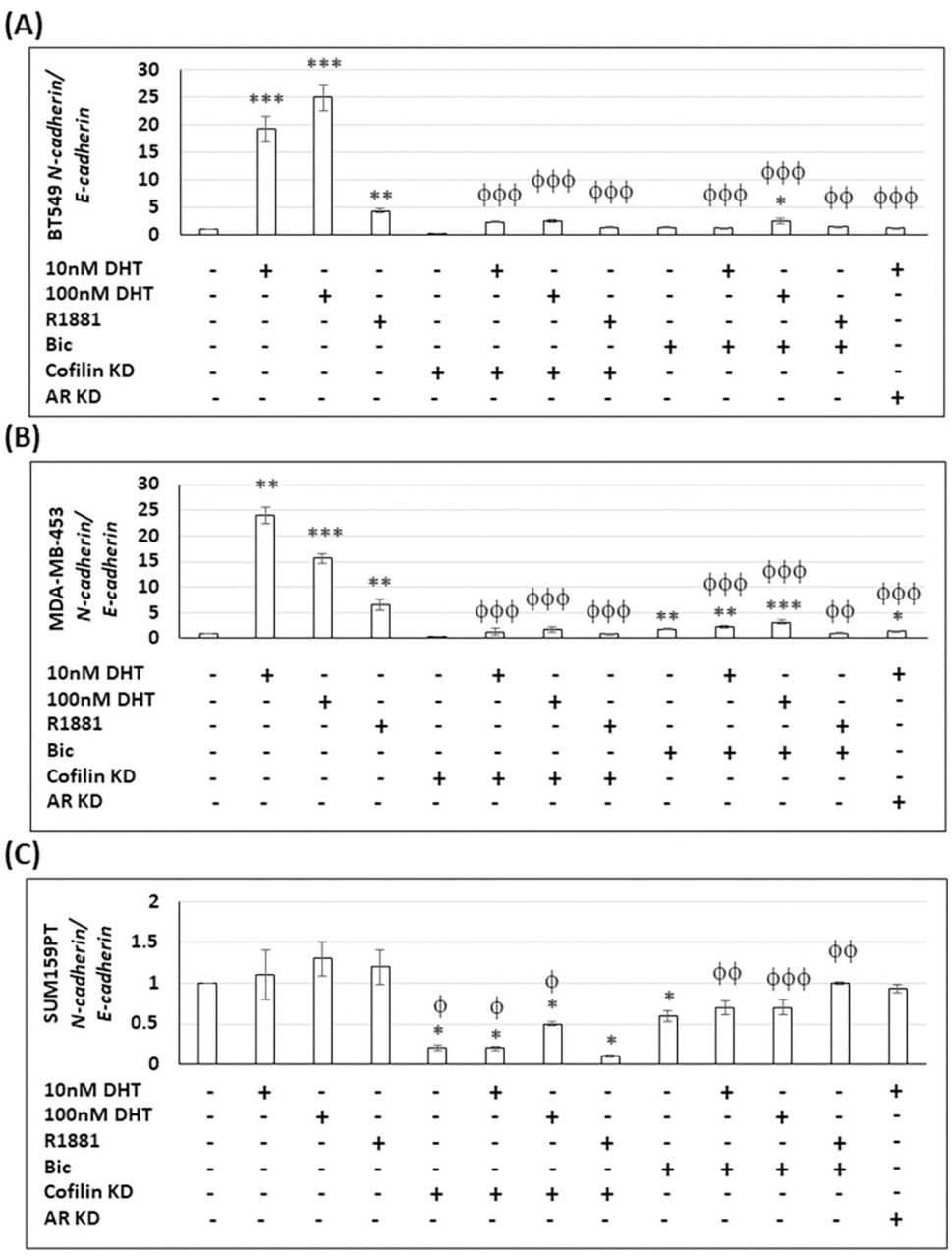

**Fig 8. Expressing cofilin shRNA negates androgen effects on *N-cadherin/E-cadherin* mRNA in TNBC cells.** Results shown are *N-cadherin/E-cadherin* qRT-PCR fold change as compared to vehicle-treated control cells [set as 1 arbitrary unit (a.u.)] in (A) BT549, (B) MDA-MB-453, and (C) SUM159PT cells. Note different scale on Y-axis for SUM159PT cells. Values were normalized to *GAPDH*. Three independent experiments each performed in duplicate. * $p < 0.05$, ** $p < 0.01$, *** $p < 0.001$ versus control, $\phi$ $p < 0.05$, $\phi\phi$ $p < 0.01$, $\phi\phi\phi$ $p < 0.001$ versus androgen-treated cells.

for individual cell migration [69, 88] (S10 Fig and Table 1), the latter two assays depend on chemotaxis, the directional migration of cells sensing their surroundings and following biochemical and mechanical cues.

The migration rate of BT549 cells was significantly increased (about 1.5 to 2 fold) by androgen treatment in both wound healing (Fig 9A and S9 Fig) and transwell migration (Fig 9B and S9 Fig) and this increase was largely negated by ***bicalutamide*** and AR silencing. Silencing

cofilin in BT549 cells reduced migration rates by 75–80% but failed to restore the rates of androgen-treated cells to near those of untreated BT549 cells (except for 10 nM DHT in wound healing assay), suggesting that androgens in BT549 cells can use a cofilin-independent pathway, or at least one in which low levels of cofilin are sufficient, for actin regulation during migration (Fig 9A and 9B and S9 Fig). MDA-MB-453 cells showed a similar but less dramatic response to androgens in both assays that were reduced to near control (untreated) levels by

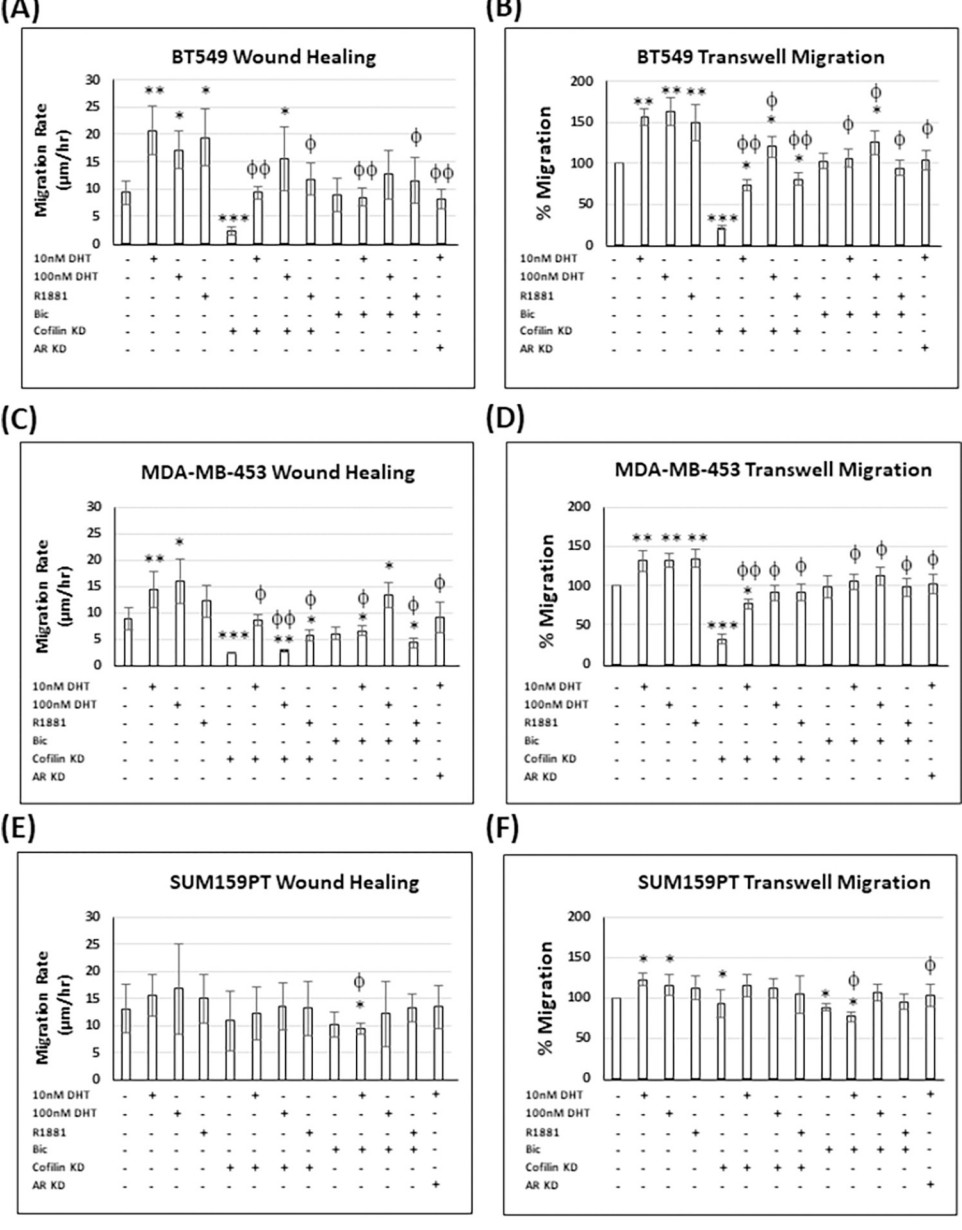

**Fig 9. Androgens increase BT549 and MDA-MB-453 wound healing and transwell cell migration rate, while knocking down cofilin reduces it in the three TNBC cell lines.** TNBC cells (A) BT549, (C) MDA-MB-453, and (E) SUM159PT were grown into monolayers and wounds were made with a sterile tip. The wound area was measured at 0 hr and later at 10–12 hr, and migration rate was expressed as μm/h. Additionally, cells [BT549 (B), MDA-MB-453 (D), and SUM159PT (F)] were serum-starved, seeded on collagen I-precoated filters and then subjected to transwell migration assay. Cell migration is expressed as percent of control cells. Three independent experiments each performed in triplicate. * $p < 0.05$, ** $p < 0.01$, *** $p < 0.001$ versus control, $\phi$ $p < 0.05$, $\phi\phi$ $p < 0.01$ versus androgen.

**Table 1. Cell migration parameters as measured by μ-slide chemotaxis assay in control and treated TNBC cell.**

*A. BT549*

| Treatment | 10% FBS L/R* | FMI[II] | FMI^ | Velocity (μm/hr) | Rayleigh Test p-value | Evidence of chemotaxis | % chemotaxing cells |
|---|---|---|---|---|---|---|---|
| Control | -/- | 0.016513 | 0.035784 | 10.77 | 0.27980 | No | 43.3% |
| Control | +/+ | -0.02507 | 0.027295 | 12.80 | 0.15098 | No | 68.0% |
| Control | +/- | -0.0832 | 0.03098 | 23.37 | 0.02945 | Yes | 73.1% |
| DHT 10nM | +/- | -0.08666 | -0.07492 | 16.71 | 0.88000 | No | 71.7% |
| DHT 100nM | +/- | 0.00404 | -0.05586 | 15.49 | 0.99182 | No | 59.1% |
| R1881 | +/- | -0.05189 | -0.08559 | 15.77 | 0.10755 | No | 69.6% |
| Cofilin KD | +/- | 0.047158 | 0.05688 | 11.80 | 0.61584 | No | 42.1% |
| DHT 10nM/KD | +/- | - | - | - | - | - | - |
| DHT 100nM/KD | +/- | - | - | - | - | - | - |
| R1881/KD | +/- | - | - | - | - | - | - |

*B. MDA-MB-459*

| Treatment | 10% FBS L/R* | FMI[II] | FMI^ | Velocity (μm/hr) | Rayleigh Test p-value | Evidence of chemotaxis | % chemotaxing cells |
|---|---|---|---|---|---|---|---|
| Control | -/- | 0.01110 | 0.03457 | 23.2 | 0.44751 | No | 46.2% |
| Control | +/+ | -0.00502 | 0.05872 | 15.97 | 0.73444 | No | 55.56% |
| Control | +/- | -0.08198 | 0.02202 | 20.62 | 0.01499 | Yes | 62.86% |
| DHT 10nM | +/- | -0.17942 | -0.06779 | 30.99 | $3.32 \times 10^{-7}$ | Yes | 81.3% |
| DHT 100nM | +/- | -0.09424 | 0.09295 | 34.43 | 0.00522 | Yes | 77.3% |
| R1881 | +/- | -0.08943 | 0.01506 | 26.80 | 0.00245 | Yes | 77.6% |
| Cofilin KD | +/- | 0.013748 | -0.01192 | 24.4 | 0.97219 | No | 46.8% |
| DHT 10nM/KD | +/- | -0.01452 | -0.00659 | 24.6 | 0.50919 | No | 48.0% |
| DHT 100nM/KD | +/- | 0.01087 | 0.00952 | 21.21 | 0.89979 | No | 47.4% |
| R1881/KD | +/- | 0.00501 | 0.00772 | 22.56 | 0.87802 | No | 50.0% |

*C. SUM159PT*

| Treatment | 10% FBS L/R* | FMI[II] | FMI^ | Velocity (μm/hr) | Rayleigh Test p-value | Evidence of chemotaxis | % chemotaxing cells |
|---|---|---|---|---|---|---|---|
| Control | -/- | 0.02182 | 0.03561 | 10.37 | 0.78219 | No | 41.2% |
| Control | +/+ | -0.00632 | 0.01914 | 14.35 | 0.85837 | No | 59.4% |
| Control | +/- | -0.08831 | 0.02184 | 16.50 | $4.5753 \times 10^{-5}$ | *Yes* | 76.4% |
| DHT 10nM | +/- | -0.07534 | -0.00826 | 17.25 | $2.53 \times 10^{-5}$ | Yes | 80.0% |
| DHT 100nM | +/- | -0.20211 | 0.10789 | 20.80 | $2.25 \times 10^{-6}$ | Yes | 83.3% |
| R1881 | +/- | -0.07079 | -0.02408 | 21.79 | 0.01485 | Yes | 75.0% |
| Cofilin KD | +/- | -0.02547 | -0.00576 | 23.9 | 0.73018 | No | 57.5% |
| DHT 10nM/KD | +/- | -0.01796 | 0.04558 | 22.11 | 0.02767 | No | 60.0% |
| DHT 100nM/KD | +/- | -0.06902 | -0.03205 | 23.9 | 0.02518 | Yes | 70.4% |
| R1881/KD | +/- | -0.05767 | -0.04138 | 10.66 | 0.34313 | No | 60.7% |

*L: Left chamber; R: Right chamber, FMI[II]: Forward migration index parallel to chemoattract gradient, FMI^: Forward migration index perpendicular to chemoattract gradient.

AR silencing and bicalutamide treatment (Fig 9C and 9D and S9 Fig). Cofilin KD in MDA-MB-453 cells also strongly reduced the migration rates and was able to restore these rates to near control levels in androgen-treated cells (except for the 100 nM DHT treatment in the wound healing assay). As with other assays, the SUM159PT cells showed minimally significant effects on migration with androgens whether or not cofilin was silenced (Fig 9E and 9F and S9 Fig).

The chemotaxis μ-slide allows for long-term analysis of chemotaxis as well as the observation of slow migrating mammalian cells [69]. Initially, epidermal growth factor (EGF, 10 nM)

[89] was used as the chemoattractant, and the EGF concentration profiles (using an estimated diffusion coefficient of D = 1 x $10^{-6}$ cm$^2$/s for EGF in buffer) in the central chamber were analyzed (S10A Fig). Since the EGF diffusion front reaches the middle of the central chamber about six hr after EGF addition to the left reservoir, cell trajectories were followed starting at t = 6 hr. Surprisingly, the rat mammary adenocarcinoma MTLn3 cells as well as the three TNBC cell lines studied were not attracted to EGF (S10B Fig). Thus, we switched to growth factor rich 10% FBS as the chemoattractant (S10C Fig) [69]. Each of the TNBC cell lines responded positively to FBS (S10C Fig; Table 1). Surprisingly, androgen-treated BT549 cells did not exhibit chemotaxis to FBS (Table 1) whereas untreated BT549 cells did. Furthermore, the migration rate of androgen-treated BT549 cells (16.0 µm/hr averaged over all androgens) was reduced from that of control cells (23.4 µm/hr). Cofilin KD BT549 cells did not exhibit chemotaxis and had a slower rate of migration (11.8 µm/hr) (Table 1).

MDA-MB-453 cells also exhibited chemotaxis to FBS (S10C Fig) but unlike BT549 cells, androgens stimulated migration rates by 30–60% (Table 1). MDA-MB-453 cofilin KD cells had similar velocities to those of untreated control, but chemotaxis was inhibited, and androgen addition had no stimulatory effect on their velocity or chemotaxis (Table 1). SUM159PT cells demonstrated a chemotaxic response to serum (S10C Fig) with an average velocity (16.5 µm/hr) about 75% of the velocities of the other two cell lines. Androgen addition increased the velocity by about 20% while not affecting chemotaxis. SUM159PT cofilin KD cells did not exhibit chemotaxis, but migration rates increased although they remained unaffected by DHT addition. However, a striking decline in migration rate (10.7 µm/hr) of cofilin KD SUM159PT cells occurred upon R1881 treatment (Table 1). This was unexpected since R1881 in both control and cofilin KD SUM159 cells had very little effect on migration rates in wound healing or transwell assays. 100 nM DHT restored chemotaxis to cofilin KD SUM159PT cells (Table 1).

## Discussion

The current study was undertaken to examine, in three triple negative breast cancer cell lines of different molecular subtypes, the role of the androgen receptor signaling pathway in the phospho-regulation of ADF/cofilin proteins, the major protein family regulating actin dynamics during cell migration [36, 90]. Cofilin-1 is the only member of the ADF/cofilin family expressed in each of the cell lines and is more highly phosphorylated (inactivated in actin binding) in BT549 and MDA-MB-453 cells than in the SUM159PT cell line. Androgen treatments led to significant dephosphorylation of cofilin in BT549 and MDA-MB-453 cells but had no significant effect on phospho-cofilin levels in SUM159PT cells suggesting that major differences are likely in cell behaviors dependent on actin assembly dynamics between the SUM159 and BT549/MDA-MB-453 cell lines, which were then explored in more detail.

Several upstream factors involved in cofilin phospho-regulation were analyzed in each cell line in the absence and presence of androgens (Fig 3). Total Rac1, chronophin and slingshot phosphatases, but not total LIMK1 or its active phosphorylated form, were upregulated 2-8-fold by androgens in BT549 and MDA-MB-453 cells, suggesting androgen-induced cofilin activation worked via increased cofilin phosphatase levels in these cell lines. SUM159PT cells showed no increase in cofilin phosphatases but had a 1.5-3-fold androgen-induced increase in total RhoA and total Rac1 and a 2-6-fold increase in total LIMK1. Surprisingly, in spite of the large increase in LIMK1, there was an androgen-induced decline in the pool of the active LIMK1 (phospho-Thr508) [70] in SUM159PT cells. Although we did not quantify the amount of active PAKs, which are one family of Rac1-activated kinases for LIMK1, there was an androgen-induced decrease in the other direct activator of LIMK1, Rho kinase (ROCK), suggesting

the lack of LIMK1 activation is a decline in its upstream activators. Furthermore, we suggest that this mechanism is manifested locally within the membrane proximal region of the leading edge of polarized SUM159PT cells where we see no androgen-induced accumulation of a band of phosphorylated cofilin that is prominent within 150 nm of the membrane in both BT549 and MDA-MB-453 cells (Fig 4I). Androgen-induced increases in phospho-LIMK1 occur in BT549 and MDA-MB-453 cells but not in SUM159PT cells. Furthermore, the proximity of microtubule +tips in the region behind the leading edge of BT549 (and MDA-MB-453 cells), but not SUM159PT cells, might be a delivery mechanism for cofilin phosphatases such as SSH-1L. SSH-1L phosphatase requires binding to F-actin for its ability to dephosphorylate cofilin but it also can dephosphorylate and inactivate LIMK1 [42, 91] to help maintain the gradient of cofilin activity within the extending lamellipodium.

Several proliferation and behavioral assays were then performed to examine the differences among the cell lines comparing responses in both control cells and those in which either cofilin or androgen receptor was silenced. Two androgens were used, the natural androgen dihydro-testosterone (DHT), and R1881 (methyltrienolone), a biologically potent synthetic androgen that is minimally metabolized compared to DHT. R1881 has a higher affinity to AR than DHT, however, its dissociation rate is faster (68 h versus 111 h, respectively) [92]. To simplify the comparisons for the reader, each of the Figs (Figs 6–9 and S3, S4, S6 and S8 Figs) has results from BT549 cells on top, MDA-MB-453 cells in the middle and SUM159PT cells on the bottom. Columns start with control (untreated cells) on the left, followed by two DHT concentrations (10 nM and 100 nM) and R1881. This pattern is repeated in the middle 4 columns in cofilin KD cells, and then again in control cells in the presence of the AR antagonist bicaluta-mide. In some cases, an additional column is added with AR knockdown to further confirm the response is AR-dependent. Although some responses were of high statistical significance and others marginally significant or just trending in that direction, the patterns of the behavior observed by quickly perusing the column profiles for each result gives a clear pattern of behavior similarities and differences among TNBC lines.

Proliferation rates of all three cell lines were stimulated 20% to 80% by androgens at each time point (24, 48 and 72 hr) (S3 Fig). The enhanced cell proliferation brought about by androgen treatment could be due to the inhibitory effects of activated AR on the tumor suppressor proteins p73 and p21 in TNBC cells [79]. Reduction of growth rate to control levels by addition of bicalutamide suggests growth stimulation is AR mediated. It has been reported that bicalutamide reduces cell proliferation in androgen treated TNBC cells by causing cell cycle arrest at the G0/G1 phase as well as reducing AR protein level [93]. S4 Fig shows that DHT at 10 and 100 nM significantly reduced the multinucleated phenotype (assessed after 72 h of addition) in MDA-MB-454 and SUM159PT cells but not in BT549 cells, whereas R1881 significantly reduced multinucleation in BT549 and SUM159PT cells but not in MDA-MB-453 cells, suggesting a faster dissociation rate of R1881/AR complex in MDA-MB-453 as compared to the other cell lines. Expression of the cofilin hnRNA silencing vector for 48 hr reduced cofilin expression to about 25% of starting value, and to about 10% by 72 hr, in keeping with a 50% decline for each 24 hr period as reported previously [60]. Growth rates in each TNBC cell line with cofilin KD declined over time with an increase in the multinucleated phenotype as expected because of the requirement for cofilin in cytokinesis. Androgen treatment of the three TNBC cell lines failed to recover cofilin-mediated growth decline, suggesting stimulation of a cofilin-independent pathway for this enhanced proliferation.

Initiation of cell polarity in epithelial cells is mediated by reducing a membrane proximal layer of F-actin which restricts membrane protrusion through its binding to membrane proteins [94]. The reduction of membrane proximal F-actin arises from an asymmetric distribution of active SSH and LIMK, enhancing cofilin activity at the cell membrane for what will

eventually become the leading edge of the cell. TNBC cells responded to androgen treatment by increasing the polarized cell phenotype (Fig 6), and BT549 and MDA-MB-453 cell length/breadth ratio (S6 Fig). The ability of bicalutamide and AR silencing to reduce the polar phenotype to control levels, suggests that polarity induction is AR dependent. Cofilin KD also reduced the polar phenotype in control TNBC cells well below control levels and restored the androgen-mediated enhancement in MDA-MB-453 cells to control level. This is turn suggests the androgen effects are at least partially dependent on cofilin spatial regulation in this cell line.

Focal adhesion area and cell adhesion in general are inhibited by androgens in BT549 and MDA-MB-453 cells and this inhibition is AR-dependent. SUM159PT cells show no decrease in focal adhesion area with androgen treatment nor any effect of the bicalutamide or AR silencing, although cofilin KD increases both focal adhesion area and cell adhesion similarly to what occurs in both BT549 and MDA-MB-453 cells (Fig 7). These differences between BT549/MDA-MB-453 and SUM159PT cell lines suggest that the *N-cadherin/E-cadherin* ratio might differ between these cell lines. Indeed, by normalizing the ratio in each line to 1:1 in untreated (control) cells, no effect of androgen treatment on the ratio occurs in the SUM159PT line, but a very large 5 to 25-fold increases in this ratio are found in BT549 and MDA-MB-453 cells, which are reduced to near control level with bicalutamide and AR KD (Fig 8). Only BT549 cells show an androgen-dependent reduction in adhesion in cofilin KD cells (Fig 7) which is accompanied by an androgen-induced increase in the *N/E cadherin* ratio (Fig 8). Loss of E-cadherin expression is correlated with weaker cell-substrate adhesion [95]. However, change in the *N/E cadherin* ratio can partly explain the enhanced spreading and adhesion induced by androgen treatment of cofilin KD MDA-MB-453 and SUM159PT cells (Fig 7). Furthermore, androgen treatment did not alter degradation area around invadopodia except for an increased degradation at the highest level of DHT in SUM159PT cells. As expected, cofilin KD in all cell lines reduced the degradation area around invadopodia and androgen addition did not reverse the effect (S8 Fig).

The migration rates of BT549 and MDA-MB-453 cells are significantly increased by androgens in both wound healing and transwell migration assays (Fig 9 and S9 Fig), and in the chemotaxis μ-slide assay for MDA-MB-453 cells (Table 1). The migration of both cell lines in response to androgens is negated by AR silencing and bicalutamide treatment, in addition to cofilin KD in MDA-MB-453 cells. SUM159PT cells are unaffected in migration assays by androgens, whether cofilin is silenced or not. The positive migration response of BT549 and MDA-MB-453 cells to androgens, and the lack of response of SUM159 cells, can be explained by both the increased pool of active cofilin and its spatial distribution (removed from the 150 nm closest to the plasma membrane of the leading edge as determined from an increased p-cofilin/total cofilin ratio; Fig 4).

The spatial regulation of cofilin activity at the leading edge of migratory cells is essential for cell protrusion, establishment of polarization, motility, and invasion. Dephosphorylated active cofilin is present toward the rear of the leading edge where it helps depolymerize actin filaments to recycle the monomers to maintain filament assembly at the membrane and maintain treadmilling of filaments. The Arp2/3 complex nucleates assembly of actin filaments at the membrane where active cofilin would have a negative impact on this process [96]. This specific spatial localization of p-cofilin is due to LIMK1 recruitment to the plasma membrane through palmitoylation of its N-terminal Cys7 and Cys8; this lipid addition not only controls LIMK1 localization but also its activation by membrane bound Rac1 and PAK, a major LIMK1 activator [97, 98]. In fully polarized BT549 (and MDA-MB-453) cells, the high ratio of p-cofilin/total cofilin is localized distal to microtubule tips at the leading edge (Fig 5H). Many microtubules are oriented perpendicular to the direction of migration as compared to very few in non-

polarized cells (Fig 5D and 5H). The exact role of microtubules in helping to maintain the distribution of inactive/active cofilin, through delivering cofilin phosphorylation signaling molecules or restricting the delivery of cofilin dephosphorylation signaling molecules, needs further analysis.

Androgen treatment induced BT549 transwell migration (Fig 9B), a chemotaxis assay, and failed to do so in the chemotaxis μ-slide assay (Table 1). Since these cells cannot be observed under a microscope as they migrate through the transwell in order to examine their behavior, their enhanced migration could have been caused by random rather than guided migration, a flaw in this assay [88]. In BT549 cells, androgens only stimulated the collective migration of cells (wound healing assay; Fig 9A), but not chemotaxis when followed by time-lapse microscopy (Table 1). Collectively migrating cancer cells have been found to be more aggressive and show chemoresistance and thus more clinically relevant than individually migrating cells [99].

In contrast to BT549 and MDA-MB-453, androgen treatment did not stimulate SUM159PT migration (Fig 9C and Table 1). SUM159PT cells showed an androgen-induced increase in area that was reduced to control in AR KD cells and by bicalutamide in R1881-treated cells but not in DHT-treated cells suggesting bicalutamide is able inactivate the R1881/AR complex more efficiently than the DHT/AR complex. Cofilin KD SUM159PT cells had an increased area over control cells with or without androgen treatment. The large cell to cell variability in SUM159PT cell length/breadth ratio resulted in few significant differences across treatments (S6 Fig). The lack of response of SUM159 cells to androgens was consistent among the different processes studied except increasing the polarized cell phenotype (Fig 6). Androgen treatment was effective in restoring the polarized phenotype above control levels in cofilin KD SUM159PT cells, a somewhat surprising finding given the apparent necessity for cofilin in breakdown of membrane proximal F-actin, suggesting that what looks like developing lamellipodia in control and cofilin KD SUM159PT cells, both of which are missing the membrane perpendicular microtubules, are not organized in a manner consistent with normal polarized migration (Fig 5I–5P).

In summary, androgen treatment of MDA-MB-453 cells and to a lesser extent in BT549 cells led to enhanced cell proliferation, increased uninucleated cells, actin reorganization, cell polarization, active and inactive cofilin redistribution at the leading edge, enhanced *N-/E-cadherin* mRNA ratio, and reduced adhesion, together resulting in increased migration rates. Bicalutamide treatment of androgen-treated BT549 and MDA-MB-453 cells as well as knocking down AR in 10 nM DHT-treated cells were able to inhibit androgen-enhanced effects demonstrating that these processes are AR-dependent. Suppression of cofilin in BT549 cells negated androgen-mediated reduction in cell adhesion and *N/E cadherin* ratio, while in MDA-MB-453 it negated androgen-enhanced cell polarization and cell migration (wound healing and transwell) and androgen-mediated decline in N/E cadherin ratio, suggesting that cofilin regulation downstream of activated AR is not only cell line-specific, but it also depends on which actin-mediated process was analyzed. There are several known signaling pathways that could link AR with cofilin ser3 phospho-regulation. Androgen effects in these cells could be mediated by Rac GTPase, which enhances ROS production by NADPH-oxidase and in turn activates SSH-1L phosphatase [100]. SSH-1L in turn can inhibit LIMK1, thus activating cofilin [42]. However, there are several other modes of regulation of cofilin including oxidation, ubiquitinylation, NEDDylation, and O-GlucNAcylation as well as a host of other proteins that can influence the ability of cofilin to bind to actin [36]. Although MDA-MB-453 cells demonstrated some control of cofilin through an AR-dependent mechanism, other AR-dependent pathways need to be further investigated. Non-cofilin-dependent mechanisms that modulate migration of SUM159PT cells need to be investigated.

## Supporting information

**S1 Data.**
(ZIP)

**S1 Raw images.**
(PDF)

**S1 Fig. Representative 2D western blots of extracted control and androgen-treated TNBC proteins immunolabeled with rabbit polyclonal antibody (1439) that recognizes ADF, phospho-ADF, cofilin, and phospho-cofilin with equal sensitivity.**
(TIF)

**S2 Fig. Maximum knock down of cofilin or AR was achieved by 72 hr post-treatment.** (A) Representative western blots of lysates from cells infected with adenovirus expressing control non-silencing shRNA (C.shRNA) and cells infected with adenovirus expressing cofilin shRNA. The experiment was repeated three times and the corresponding quantification is shown in (C). (B) Representative western blots of lysates from cells transfected with a control non-silencing siRNA (C.siRNA), and cells transfected with silencing siRNA against AR. The experiment was repeated three times and the corresponding quantification is shown in (C). (C) Quantification of cofilin and AR protein levels in shRNA/siRNA infected/transfected-TNBC cells relative to their levels in C.shRNA-infected or C.siRNA-transfected cells (set as 1.0 a.u.) obtained from densitometers of immunoblots. (D) Representative western blots of lysates from cells infected for 72 hr with adenovirus for expressing cofilin shRNA (cofilin KD; KD in the Fig). Blots were immunolabeled using the AR antibody and a monoclonal antibody to GAPDH as a loading control. The experiment was repeated three times and the corresponding quantification is shown in (E). (E) Quantification of AR levels in cofilin KD cells as compared to control uninfected cells (set as 1.0) obtained from densitometers of immunoblots.
(TIF)

**S3 Fig. Knocking down cofilin in the presence or absence of androgens inhibits cell proliferation of TNBC cells.** TNBC cells (A) BT549, (B) MDA-MB-453, and (C) SUM159PT were treated for 24, 48, and 72 hr and the growth percentage (Growth %) relative to the vehicle-treated control cells (set as 100%) was measured by MTT assay. Bars = mean ± SEM of three independent experiments performed in triplicates. * $p < 0.05$, ** $p < 0.01$, *** $p < 0.001$ versus control, ᵠ $p < 0.05$, ᵠᵠ $p < 0.01$, ᵠᵠᵠ $p < 0.001$ versus androgen-treated cells.
(TIF)

**S4 Fig. Cofilin depletion in control and androgen-treated TNBC cells increased percentage of multinucleation.** TNBC cells (A) BT549, (B) MDA-MB-453, and (C) SUM159PT were stained with DAPI and scored for the percentage of uninucleation and multinucleation. $n \geq 100$ cells in each experiment, three independent experiments. * $p < 0.05$, ** $p < 0.01$, *** $p < 0.001$ versus control, ᵠ $p < 0.05$, ᵠᵠ $p < 0.01$, ᵠᵠᵠ $P < 0.001$ versus androgen-treated cells.
(TIF)

**S5 Fig. Knocking down cofilin in TNBC cells causes cell rounding and stress fiber formation while androgen treatment causes cell elongation.** TNBC cells were stained with fluorescent-conjugated phalloidin. (A) Apolar BT549 cell displaying cuboidal epithelium morphology with cortical actin cytoskeleton. (B) Crescent polar BT549 cell. (C) Kite-like polar BT549 cell. (D) Cofilin shRNA-transfected BT549 cell showing thick stress fibers and cell rounding. (E) Androgen-treated BT549 cell showing a dramatic change in cell morphology with an elongated cytoplasmic extension that resembles a neurite. (F) Bipolar BT549 cell

having two lamellipodia. (G) MDA-MB-453 cells grow as clusters (grape-like or stellate structures). (H) Individual MDA-MB-453 cell showing prominent cortical actin. (I) Crescent polar MDA-MB-453 cell. (J) Kite-like polar MDA-MB-453 cell. (K) Knocking down cofilin in MDA-MB-453 cells resulted in the formation of thick stress filaments and cell rounding (apolar fried-egg morphology). (L) SUM159PT cells grow mainly in groups. (M) Individual SUM159PT cell showing homogeneously distributed actin filament and prominent lamellipodium. (N) Crescent polar SUM159PT cell. (O) Kite-like polar SUM159PT cell. (P) SUM159PT cell expressing cofilin shRNA shows cell expansion and prominent actin stress fibers. Scale bar: 10 μm. (Q) Androgen-treated TNBC cells exhibit the kite-like morphology. Cells were stained with fluorescent-conjugated phalloidin and the percentage of polarized cells exhibiting the kite-like morphology, or the crescent morphology was scored. Data are expressed as mean ± SEM, n ≥ 100 cells in each experiment, three independent experiments. $^*$ p<0.05, $^{**}$ p< 0.01 versus control.
(TIF)

**S6 Fig. Effects of cofilin silencing with and without androgen treatment on TNBC cell area and length/breadth ratio.** Control and treated TNBC cells were stained with fluorescent-conjugated phalloidin and cell area, length (the span of the longest chord through the cell), and breadth (the caliper width of the cell perpendicular to the longest chord) were measured by Metamorph software (Molecular Devices). (A) BT549 cell area ($\mu m^2$). (B) BT549 length/breadth ratio. (C) MDA-MB-453 cell area ($\mu m^2$). (D) MDA-MB-453 length/breadth ratio. (E) SUM159PT cell area ($\mu m^2$). (F) SUM159PT length/breadth ratio. Data are expressed as mean ± SEM, n ≥ 100 cells in each experiment, three independent experiments. $^*$ p <0.05, $^{**}$ p <0.01 versus control, $^\phi$ p <0.05, $^{\phi\phi}$ p <0.01, $^{\phi\phi\phi}$ p <0.001 versus androgen-treated cells.
(TIF)

**S7 Fig. Percentage of change (% change) of focal adhesion in TNBC-treated cells compared to control cells (set as 100%).**
(TIF)

**S8 Fig. Androgens do not affect the extracellular matrix degradation activities of TNBC cells, while suppression cofilin expression inhibits it.** (A) Cells were cultured on fluorescent gelatin-coated cover slips and stained with fluorescent-phalloidin to visualize invadopodia (arrows). Scale bar: 10 μm. The degradation area of gelatin ($\mu m^2$) was quantified in Metamorph and divided by number of cells in the same field and expressed as degradation area ($\mu m^2$)/cell in (B) BT549 cells, (C) MDA-MB-453 cells, and (D) SUM159PT cells. n ≥ 25 cells in each experiment, three independent experiments. $^*$ p <0.05, $^{**}$ p < 0.01 versus control, $^\phi$ p < 0.05, $^{\phi\phi}$ p <0.01, $^{\phi\phi\phi}$ p <0.001 versus androgen-treated cells.
(TIF)

**S9 Fig. Percentage of change (% change) of migration rate in wound healing and transwell migration assays in TNBC-treated cells compared to control cells (set as 100%).**
(TIF)

**S10 Fig. Chemotaxis analysis of TNBC cells.** (A) EGF concentration profiles in the central chemotaxis chamber over time. For an initial concentration of 10 nM EGF in the left reservoir, it takes approximately 6 hr for the EGF diffusion front to cross the midpoint of the central chamber. The EGF concentration profile becomes nearly linear by 48 hr. (B) Control untreated cells were seeded on collagen IV-treated chemotaxis μ-slide with 10 nM EGF in the left reservoir and chemoattractant-free media in the right reservoir (EGF +/-). (C) Control and treated cells were seeded on collagen IV-coated chemotaxis μ-slide. 10% FBS -/-: negative/negative

control cells with no chemoattractant in either reservoir, 10% FBS +/+: positive/positive control cells with 10% FBS in both reservoirs, 10% FBS +/-: positive/negative control or treated cells with 10% FBS in the left reservoir. Cell tracking was performed in MATLAB using the image processing software CellTracker. Cell trajectories were then imported into the ImageJ software plugin "Chemotaxis and Migration Tool" and extrapolated to (x,y) = (0,0) at time 6 hr, where the x-axis is parallel to the chemoattractant gradient and the y-axis is perpendicular to the gradient. On average 30–60 cells were tracked per experiment, and each experiment was repeated three times.
(TIF)

## Acknowledgments

We gratefully acknowledge helpful discussions and/or technical assistance from Ms. Laurie S. Minamide, Dr. O'Neil Wiggan, and Dr. Thomas Kuhn. The assistance of Andrew Tonsager in running RT-PCR is greatly appreciated. The manuscript is dedicated to the memory of Ghalib Al-Nusair.

## Author Contributions

**Conceptualization:** Lubna Tahtamouni, James Bamburg.

**Data curation:** Lubna Tahtamouni, Mamoun Ahram, Ashok Prasad.

**Formal analysis:** Lubna Tahtamouni, Ashok Prasad, James Bamburg.

**Funding acquisition:** Lubna Tahtamouni.

**Investigation:** Lubna Tahtamouni, Ahmad Alzghoul, Sydney Alderfer, Jiangyu Sun.

**Project administration:** Lubna Tahtamouni.

**Supervision:** Lubna Tahtamouni, James Bamburg.

**Writing – original draft:** Lubna Tahtamouni, Mamoun Ahram.

**Writing – review & editing:** Lubna Tahtamouni, James Bamburg.

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
