## [Decision Letter · Decision Letter 0]

14 Oct 2022

PONE-D-22-26107Different androgen receptor-mediated pathways modulate cofilin phospho-regulation to control polarized migration of triple negative breast cancer lines of different molecular subtypesPLOS ONE

Dear Dr. Tahtamouni,

Thank you for submitting your manuscript to PLOS ONE. After careful consideration, we feel that it has merit but does not fully meet PLOS ONE’s publication criteria as it currently stands. Therefore, we invite you to submit a revised version of the manuscript that addresses the points raised during the review process. As indicated below, the reviewers have different opinions. As indicated by my comments, while there is a substantial body of data, it does not support the conclusions reached that coffin mediates androgen receptor activity in the TNBC cells studied. The manuscript is not therefore acceptable for publication in PLOS One. You are invited to address the reviewer comments and submit a revision but please note that all conclusions must be supported by the data provided. 

We look forward to receiving your revised manuscript.

Kind regards,

Ivan R. Nabi, Ph.D.

Academic Editor

PLOS ONE

5. Please include captions for your Supporting Information Tables at the end of your manuscript, and update any in-text citations to match accordingly. Please see our Supporting Information guidelines for more information: http://journals.plos.org/plosone/s/supporting-information.

Additional Editor Comments :

This is a comprehensive analysis of the effect of androgen stimulation on the migratory phenotype of three TNBC cell lines, and specifically its effect on cofilin modulation of the actin cytoskeleton. In general, there is a large amount of data that, while done appropriately, could be more clearly presented. Of major concern is the fact that the data does not support the conclusion of the authors that :

"Different androgen receptor-mediated pathways modulate cofilin phospho-regulation to control polarized migration of triple negative breast cancer lines of different molecular subtypes"

"Suppression of cofilin in MDA and to a lesser extend BT cells abrogated the androgen effects suggesting cofilin regulation is involved."

The only data that I was able to identify that supported a role for cofilin downstream of androgen receptor activation was changes in the N-cadherin/E-cadherin ratio that weren't followed up on in the paper. Most other effects of AR activation were cofilin independent and suggest that AR uses a cofilin-independent pathway to promote migration, as indicated at numerous places in the text by the authors. Inpaticualr, the data provided do not support the summary figure 10. As such, conclusions reached that cofilin mediates AR effects on TNBC cell migratory phenotypes are not supported by the data and the manuscript as such is not acceptable at PLOS One. Conclusions reached for a revised submission must be supported by the data provided and reflect the data obtained. I also suggest that the authors review all the data and present the data in a more concise and easy to follow manner that supports the conclusions drawn.

Comments:

1. Use of abbreviations for the cell lines used in the text and figures is not appropriate and full names of the cell line names must be used at all times.

2. Inclusion of fluorescent images without quantification is not acceptable (Fig 5, S4). Where quantified clear images should be provided showing the effect measured. For instance images of the AR effect on p-cofilin at the leading edge are not shown in figure 4.

3. The rational to include data in Table format is not clear and the extensive data provided in the tables should be presented as graphs.

Reviewers' comments:

Reviewer's Responses to Questions

**Comments to the Author**

1. Is the manuscript technically sound, and do the data support the conclusions?

Reviewer #1: Yes

Reviewer #2: Partly

2. Has the statistical analysis been performed appropriately and rigorously? 

Reviewer #1: Yes

Reviewer #2: Yes

3. Have the authors made all data underlying the findings in their manuscript fully available?

Reviewer #1: Yes

Reviewer #2: Yes

4. Is the manuscript presented in an intelligible fashion and written in standard English?

Reviewer #1: Yes

Reviewer #2: No

5. Review Comments to the Author

Reviewer #1: The readability and scientific structure/format of this manuscript is at least above average. It was not difficult to read/understand and many of the concepts were explained clearly and I did not find this review assignment any more difficult than if it was from another journal.

The manuscript and its findings used methods that were technically sound and used by many other cancer cell biologists in the field. The main value of this manuscript is for investigators in the field that are also looking at levels of very specific proteins (AR +/- DHT, p-cofilin) in the breast cancer cell lines chosen for analysis. In my opinion, its these kinds of reports that provide independent sources of external validation beyond meaning. I contend that there is no significant correlation between cofilin (p or not) and DHT activation amongst these cell lines but I do see value in these results being used to support other investigator's pursuits if they so happen to be looking for "...a paper that looked at the levels of p-cofilin with DHT activation and its effects on micro-actin tubules or invadopodia...".

Reviewer #2: The current manuscript investigates the potential relationship between AR stimulation and cofilin activity. Through AR stimulation and cofilin knock downs, the authors have assessed the role of AR in cofilin modulation and migration. The status and level of expression of other motility regulators have been evaluated.

Although there is a large body of work presented, there is no clear or specific connection demonstrated between AR and cofilin modulation. The observations are in fact expected for the most part. AR stimulates motility in the TNBC lines, therefore cytoskeletal reorganization such as cofilin dephosphorylation is expected. The same is true for proliferation. A cofilin KD would impair cell division. This is not specific to AR or any other growth or migration promoting agent.

There is also no data supporting the model proposed as it has not been investigated in this manuscript. The manuscript would also benefit from editing as it is too long and reads like a thesis.

6. PLOS authors have the option to publish the peer review history of their article (what does this mean?). If published, this will include your full peer review and any attached files.

Reviewer #1: **Yes: **Hon Sing Leong

Reviewer #2: No

---

## [Author Response · Author response to Decision Letter 0]

22 Nov 2022

Editor’s comments:

1. Thank you for submitting your manuscript to PLOS ONE. After careful consideration, we feel that it has merit but does not fully meet PLOS ONE’s publication criteria as it currently stands. As indicated by my comments, while there is a substantial body of data, it does not support the conclusions reached that coffin mediates androgen receptor activity in the TNBC cells studied. The manuscript is not therefore acceptable for publication in PLOS One. You are invited to address the reviewer comments and submit a revision but please note that all conclusions must be supported by the data provided. 

Reply: We thank the editor for the encouraging words and for the critical evaluation of the manuscript. We have re-analyzed all data, again and again, and we agree that some of the conclusions reached in the original submitted manuscript were not supported by the data since we grouped two cell lines (BT549 and MDA-MB-453) in the same conclusions although their behaviors were different. 

We have revised the analysis, and thus the discussion and conclusions, and that made us re-think the original Title of the manuscript since it is not supported by the results after this careful re-analysis. The revised manuscript now has a Revised Title as well. 

Reply: The revised version of the manuscript meets PLOS ONE’S style requirements in terms of formatting of the title, authors, and main text

Reply: We thank the editor for the comment. We provided the grants numbers that supported our work. 

4. We note that you have indicated that data from this study are available upon request. PLOS only allows data to be available upon request if there are legal or ethical restrictions on sharing data publicly. For more information on unacceptable data access restrictions, please see http://journals.plos.org/plosone/s/data-availability#loc-unacceptable-data-access-restrictions. In your revised cover letter, please address the following prompts:

Reply: We apologize for not reading PLOS ONE data availability section more carefully when we submitted the original manuscript. We provided the results data as Supporting Information files (Excel Spreadsheets). 

5. PLOS ONE now requires that authors provide the original uncropped and unadjusted images underlying all blot or gel results reported in a submission’s figures or Supporting Information files. This policy and the journal’s other requirements for blot/gel reporting and figure preparation are described in detail at https://journals.plos.org/plosone/s/figures#loc-blot-and-gel-reporting-requirements and https://journals.plos.org/plosone/s/figures#loc-preparing-figures-from-image-files. When you submit your revised manuscript, please ensure that your figures adhere fully to these guidelines and provide the original underlying images for all blot or gel data reported in your submission. See the following link for instructions on providing the original image data: https://journals.plos.org/plosone/s/figures#loc-original-images-for-blots-and-gels. In your cover letter, please note whether your blot/gel image data are in Supporting Information or posted at a public data repository, provide the repository URL if relevant, and provide specific details as to which raw blot/gel images, if any, are not available. Email us at plosone@plos.org if you have any questions.

Reply: Again, we apologize for not reading PLOS ONE regulations regarding uncropped blot results more carefully when we submitted the original manuscript. We provided the results data as Supporting Information files (PDF file named S1_raw_images). 

6. Please include captions for your Supporting Information Tables at the end of your manuscript, and update any in-text citations to match accordingly. Please see our Supporting Information guidelines for more information: http://journals.plos.org/plosone/s/supporting-information.

Reply: Thank you for the comment. Both Supporting Information Tables are now Supporting Information Figures (S7 and S9 Figs). 

7. This is a comprehensive analysis of the effect of androgen stimulation on the migratory phenotype of three TNBC cell lines, and specifically its effect on cofilin modulation of the actin cytoskeleton. In general, there is a large amount of data that, while done appropriately, could be more clearly presented. Of major concern is the fact that the data does not support the conclusion of the authors that :

"Different androgen receptor-mediated pathways modulate cofilin phospho-regulation to control polarized migration of triple negative breast cancer lines of different molecular subtypes"

"Suppression of cofilin in MDA and to a lesser extend BT cells abrogated the androgen effects suggesting cofilin regulation is involved."

The only data that I was able to identify that supported a role for cofilin downstream of androgen receptor activation was changes in the N-cadherin/E-cadherin ratio that weren't followed up on in the paper. Most other effects of AR activation were cofilin independent and suggest that AR uses a cofilin-independent pathway to promote migration, as indicated at numerous places in the text by the authors. In particular, the data provided do not support the summary figure 10. As such, conclusions reached that cofilin mediates AR effects on TNBC cell migratory phenotypes are not supported by the data and the manuscript as such is not acceptable at PLOS One. Conclusions reached for a revised submission must be supported by the data provided and reflect the data obtained. I also suggest that the authors review all the data and present the data in a more concise and easy to follow manner that supports the conclusions drawn.

Reply: After re-analyzing the results again, prompted by the editor’s and reviewer’s comments, we agree that we generalized the conclusions reached to include both MDA-MB-453 cells and BT549 cells in the same conclusion, and we generalized the results reached for the different behaviors assessed although different assays for the same behavior might have resulted in different conclusions such as cell migration. The revised manuscript clearly states the conclusions reached based on the results and analysis: cofilin and AR only work together in focal adhesion and N/E cadherin ratio in BT549 cells, and in MDA-MB-453 cells they work together in cell polarization (% of polar cells), N/E cadherin ratio and wound healing/transwell migration. 

Based on our decades of work on cofilin signaling pathway, we came up with the pathway presented in Figure 10 that is not supported by the results and conclusions reached in the current work. Thus, Figure 10 has been omitted from the revised version. 

8. Use of abbreviations for the cell lines used in the text and figures is not appropriate and full names of the cell line names must be used at all times.

Reply: We thank the reviewer for the comment. The revised version has the full name of the cell lines in the text and figures. 

9. Inclusion of fluorescent images without quantification is not acceptable (Fig 5, S4). Where quantified clear images should be provided showing the effect measured. For instance images of the AR effect on p-cofilin at the leading edge are not shown in figure 4.

Reply: Figure S4 (now figure S5) has been quantified extensively. The figure itself includes representative images to describe actin organization in control and treated cells. However, from the images of fluorescent actin, more than 100 images in each category repeated at least 3 times independently, we analyzed the percentage of polarization (Figure 6), cell area and length/width ratio (Supplement Figure 6), and percentage of polarized cells in each category (Kite vs Crescent) (Supplement Figure 5Q).

Figure 4 has been quantified as well and the quantification appears in Figure 4I, however, we apologize for the mistake in the vertical axis title in the original submitted manuscript; we missed to include “Leading Edge”, this was revised in the revised version. 

As for Figure 5, it was one of the major challenges we faced during our work since staining microtubules and p-cofilin together is not straightforward, they require different fixative solutions and different protocols available to stain microtubules use paraformaldehyde and glutaraldehyde, two fixatives that should not use with cofilin antibodies. Additionally, most published protocols include a detergent with the fixation solution (or prior to fixation) to be able to see microtubules crystal clear, a step clearly, we can’t do here. Figure 5 was an attempt to verify results (unpublished previously acquired results) from our lab and one of our collaborators’ labs showing that microtubules restrict localization of cofilin and p-cofilin, the best we can do here is a description what we see from at least 25 cells in each category, a task that was time-demanding and very tedious. 

10. The rational to include data in Table format is not clear and the extensive data provided in the tables should be presented as graphs.

Reply: We thank the editor for the comment. In the revised version of the manuscript, all tables except Table 3 (now Table 1) have been converted to graphs. Presenting results of the chemotaxis assay in a table format is standard in this field. 

Reviewers' comments:

Reviewer #1: 

1. The readability and scientific structure/format of this manuscript is at least above average. It was not difficult to read/understand and many of the concepts were explained clearly and I did not find this review assignment any more difficult than if it was from another journal.

The manuscript and its findings used methods that were technically sound and used by many other cancer cell biologists in the field. The main value of this manuscript is for investigators in the field that are also looking at levels of very specific proteins (AR +/- DHT, p-cofilin) in the breast cancer cell lines chosen for analysis. In my opinion, its these kinds of reports that provide independent sources of external validation beyond meaning. 

Reply: We thank the reviewer for their encouraging words.

I contend that there is no significant correlation between cofilin (p or not) and DHT activation amongst these cell lines but I do see value in these results being used to support other investigator's pursuits if they so happen to be looking for "...a paper that looked at the levels of p-cofilin with DHT activation and its effects on micro-actin tubules or invadopodia...".

Reply: After looking at the results and conclusions through the eyes of external reviewers (the editor’s and the reviewers’) we agree that the relation between AR and cofilin phospho-regulation is not only cell line specific, but also depends on which actin-mediated process was studied. We hope that our revised version delivers more sound conclusions. 

Reviewer #2: 

1. The current manuscript investigates the potential relationship between AR stimulation and cofilin activity. Through AR stimulation and cofilin knock downs, the authors have assessed the role of AR in cofilin modulation and migration. The status and level of expression of other motility regulators have been evaluated.

Although there is a large body of work presented, there is no clear or specific connection demonstrated between AR and cofilin modulation. The observations are in fact expected for the most part. AR stimulates motility in the TNBC lines, therefore cytoskeletal reorganization such as cofilin dephosphorylation is expected. The same is true for proliferation. A cofilin KD would impair cell division. This is not specific to AR or any other growth or migration promoting agent.

Reply: We agree with the reviewer that some of the results were expected but still they had to be done as a proof of concept and to validate the reagents used such as cofilin and AR siRNAs and the different androgens. We measured proliferation rates and % of uni- and multinucleation for this reason specifically and thus they only appeared as supplementary figures. However, we disagree with the reviewer statement (AR stimulates motility in the TNBC lines, therefore cytoskeletal reorganization such as cofilin dephosphorylation is expected). There is a plethora of actin-binding proteins that can work downstream of AR to reorganize the cytoskeleton and thus stimulate motility beside cofilin, and this is exactly what we found. Activated AR worked independently from cofilin in many of the different actin-dependent behaviors including the migration assays studied. SUM159PT cells almost never had a response to androgens, while MDA-MB-453 cells used cofilin downstream of AR in wound healing and to a lesser extent in transwell migration but not in chemotaxis. 

2. There is also no data supporting the model proposed as it has not been investigated in this manuscript. 

Reply: We agree with the reviewer, the model we presented in Figure 10 was too optimistic and was based on our years long work on cofilin-mediated changes in actin dynamics. Figure 10 was deleted from the revised version and our conclusions have been revised to match the results. Only supported conclusions were stated.

3. The manuscript would also benefit from editing as it is too long and reads like a thesis.

Reply: We thank the reviewer for the comment. We have revised the manuscript and tried to shorten it. However, we try to be as detailed as we can in describing the experimental procedures to ensure reproducibility.

---

## [Decision Letter · Decision Letter 1]

15 Dec 2022

The role of activated androgen receptor in cofilin phospho-regulation depends on the molecular subtype of TNBC cell line and actin assembly dynamics

PONE-D-22-26107R1

Dear Dr. Tahtamouni,

We’re pleased to inform you that your manuscript has been judged scientifically suitable for publication and will be formally accepted for publication once it meets all outstanding technical requirements.

Kind regards,

Ivan R. Nabi, Ph.D.

Academic Editor

PLOS ONE

Additional Editor Comments (optional):

Reviewers' comments:

Reviewer's Responses to Questions

**Comments to the Author**

1. If the authors have adequately addressed your comments raised in a previous round of review and you feel that this manuscript is now acceptable for publication, you may indicate that here to bypass the “Comments to the Author” section, enter your conflict of interest statement in the “Confidential to Editor” section, and submit your "Accept" recommendation.

Reviewer #2: All comments have been addressed

2. Is the manuscript technically sound, and do the data support the conclusions?

Reviewer #2: Yes

3. Has the statistical analysis been performed appropriately and rigorously? 

Reviewer #2: Yes

4. Have the authors made all data underlying the findings in their manuscript fully available?

Reviewer #2: Yes

5. Is the manuscript presented in an intelligible fashion and written in standard English?

Reviewer #2: Yes

6. Review Comments to the Author

Reviewer #2: (No Response)

7. PLOS authors have the option to publish the peer review history of their article (what does this mean?). If published, this will include your full peer review and any attached files.

Reviewer #2: No

---

## [Editor Report · Acceptance letter]

22 Dec 2022

PONE-D-22-26107R1 

The role of activated androgen receptor in cofilin phospho-regulation depends on the molecular subtype of TNBC cell line and actin assembly dynamics 

Dear Dr. Tahtamouni:

I'm pleased to inform you that your manuscript has been deemed suitable for publication in PLOS ONE. Congratulations! Your manuscript is now with our production department. 

Kind regards, 

on behalf of

Dr. Ivan R. Nabi 

Academic Editor

PLOS ONE